

# Exact dynamical correlations of nonlocal operators in quadratic open Fermion systems: a characteristic function approach

**Qing-Wei Wang**

School of Information Engineering, Zhejiang Ocean University,
Zhoushan, Zhejiang 316022, China
Key Laboratory of Oceanographic Big Data Mining & Application of Zhejiang Province,
Zhejiang Ocean University, Zhoushan, Zhejiang 316022, China

qingweiwang2012@163.com

## Abstract

The dynamical correlations of nonlocal operators in general quadratic open fermion systems is still a challenging problem. Here we tackle this problem by developing a new formulation of open fermion many-body systems, namely, the characteristic function approach. Illustrating the technique, we analyze a finite Kitaev chain with boundary dissipation and consider anyon-type nonlocal excitations. We give explicit formula for the Green's functions, demonstrating an asymmetric light cone induced by the anyon statistical parameter and an increasing relaxation rate with this parameter. We also analyze some other types of nonlocal operator correlations such as the full counting statistics of the charge number and the Loschmidt echo in a quench from the vacuum state. The former shows clear signature of a nonequilibrium quantum phase transition, while the later exhibits cusps at some critical times and hence demonstrates dynamical quantum phase transitions.

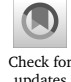
# 1 Introduction

The interaction of a quantum system with its environment [1–3] can lead to various dissipation behaviors and the emergence of new collective phenomena, such as nonequilibrium phases and phase transitions driven by dissipation [4–12], universality and dynamic scaling behaviors at quantum transitions [13–19]. Understanding and controlling the behavior of quantum dissipative systems is also fundamental to the development of quantum-enhanced cutting-edge technologies such as quantum computing [20], quantum metrology [21], quantum state preparation or quantum reservoir engineering [22–31]. Although significant experimental advancements have been made in this context [32–35], dissipative quantum many-body problems are still quite challenging in theory. Within the so-called Markovian approximation, the open systems' Liouvillian dynamics is described by the Lindblad master equation [36, 37] for the time-dependent density matrix. A standard way of analyzing the master equation is by means of perturbation methods [38–42]. In addition, some exact solutions of the nonequilibrium steady states and the full spectrum of the Liouvillian have been obtained in some specific representative cases [43–53].

One specific instance that has attracted many interests is the open fermionic systems with *quadratic Lindbladian* [54–64], which can be solved exactly. However, even for such simple solvable systems, the dynamics of nonlocal operators is still challenging and desires efficient computation methods. Here we use *nonlocal operators* to refer to those operators containing a string operator of the form $\hat{O}_j = \exp[i\phi \sum_{l \leq j} \hat{c}_l^\dagger \hat{c}_l]$ (or more generally, an exponential function of bilinear fermion operators). Such operators appear in many important physical problems. For example, string order parameters have been used to characterize topological properties of quantum systems [65–68]. They also emerge in the studies of the Tonks-Girardeau gas [69,70], the impenetrable anyons [71,72], the XY Heisenberg chain [73], and the full counting statistics of quantum transport [74, 75]. The dynamical correlation functions of nonlocal operators in dissipative systems have not been investigated systematically, even in quadratic open systems. It represents a highly nontrivial theoretical problem.

Motivated by such challenges, here we put forward a new theoretical approach to open fermion systems by applying the idea of mappings between the Liouville-Fock space $\mathcal{K}$ and a Grassmann algebra $\mathcal{G}$, which can map operators to analytic functions of Grassmann variables and vice versa. The quantum master equation is transformed to a partial differential equation of the characteristic function of the density matrix, and all physical observables can be expressed in terms of this function. We name this new approach as *characteristic function approach* since the $\mathcal{K}$-$\mathcal{G}$ mappings and the characteristic function are essential concepts. This method could be seen as a fermion analogue of the phase-space method widely used in quantum optics [76,77].

Our method, which can be useful for generic open fermion systems, is then applied to general quadratic fermion systems with linear Lindblad operators. We give exact solutions of the master equation, the steady state, the single-particle Green's function, the dynamical response function, and most importantly, the dynamical correlations of nonlocal operators. These general results are then applied to the Kitaev chain with boundary dissipation [57, 78, 79]. We obtain the spectrum of the matrix that determines the dissipative dynamics of the system, finding an excited state quantum phase transition (ESQPT) and its relationship with the nonequilibrium quantum phase transition (NQPT). We also compute the Green's functions of nonlocal excitations, namely, the hard-core anyons with statistical parameter $\phi$, and find that the propagation of the excitations displays an asymmetric light-cone for $\phi \neq 0, \pi$, and the relaxation rate increases with the statistical parameter. In addition, other types of nonlocal operator correlations such as the full counting statistics (FCS) of the charge number in a subsystem and the Loschmidt echo in quench dynamics can also be analyzed easily in our new approach and explicit formulas can be obtained. The FCS shows clear signature of the NQPT mentioned above, while the Loschmidt echo rate function exhibits cusps at some critical times in the quench from the vacuum state, giving evidence of dynamical quantum phase transitions (DQPT) in this dissipative system. These analyses demonstrate the feasibility and powerfulness of the characteristic function approach.

This paper is organized as follows. In Sec.2, we present the general formalism of the characteristic function approach and use it to give the exact solutions of various physical properties of the open fermion systems with quadratic Lindbladian, with emphasis on the dynamical correlations of nonlocal operators. In Sec.3 we analyze the boundary-driven Kitaev chain as an example, focusing on the Green's function of the hard-core anyons, the full counting statistics of the charge number in a subsystem, and the Loschmidt echo rate in a quench dynamics from the vacuum state. We conclude in Sec.4 with a summary of our main results and some discussions.

## 2 The characteristic function approach

### 2.1 Basic Formalism

We first develop a new general approach to solve quantum master equations of fermion systems. The basic idea is quite simple: the Liouville-Fock space $\mathcal{K}$ generated by fermion creation and annihilation operators $\{\hat{c}_1^\dagger, \hat{c}_1, \ldots, \hat{c}_N^\dagger, \hat{c}_N\}$ and the Grassmann algebra $\mathcal{G}$ generated by Grassmann variables $\{\bar{\xi}_1, \xi_1, \ldots, \bar{\xi}_N, \xi_N\}$ have the same dimension $2^{2N}$ and hence we can construct one-to-one mappings between these two spaces. In analogy to the phase-space functions and characteristic functions widely used in quantum optics [76], we define the mapping $\Theta$ from $\mathcal{K}$ to $\mathcal{G}$ as the *characteristic function* of the operators in $\mathcal{K}$:

$$\Theta : \hat{A} \in \mathcal{K} \rightarrow A_C(\bar{\xi}, \xi) \equiv \mathrm{Tr}[\hat{D}(\xi)\hat{A}], \tag{1}$$

where $\hat{D}(\xi) \equiv e^{\hat{c}^\dagger \xi - \bar{\xi}\hat{c}}$ is the fermion analogue of the boson displacement operator. Here we use the notations $\hat{c}^\dagger \equiv (\hat{c}_1^\dagger, \hat{c}_2^\dagger, \ldots, \hat{c}_N^\dagger)$, $\bar{\xi} \equiv (\bar{\xi}_1, \bar{\xi}_2, \ldots, \bar{\xi}_N)$, and $\hat{c} \equiv (\hat{c}_1, \hat{c}_2, \ldots, \hat{c}_N)^T$, $\xi \equiv (\xi_1, \xi_2, \ldots, \xi_N)^T$. Inversely, we have

$$\Omega : A_C(\bar{\xi}, \xi) \in \mathcal{G} \rightarrow \hat{A} = \int d\bar{\xi} d\xi \, A_C(\bar{\xi}, \xi) \left[ \frac{e^{i\pi\hat{N}} + \mathbb{1}}{2} \hat{D}^\dagger(\xi) + \frac{e^{i\pi\hat{N}} - \mathbb{1}}{2} \hat{D}(\xi) \right], \tag{2}$$

where $\hat{N} = \sum_i \hat{c}_i^\dagger \hat{c}_i$ is the total fermion number operator. It's straightforward to prove that $\Theta$ and $\Omega$ are reciprocal linear mappings. To do this, it's enough to show that for any analytic

function $f(\bar{\eta}, \eta) \in \mathcal{G}$, we have $f = \Theta[\Omega(f)]$.

$$
\begin{aligned}
\Theta[\Omega(f)] &= \int d\bar{\alpha}d\alpha f(\bar{\alpha},\alpha)\operatorname{Tr}\left[e^{i\pi\hat{N}}\hat{D}^\dagger(\alpha)\hat{D}(\eta)\right] \\
&= \int d\bar{\alpha}d\alpha f(\bar{\alpha},\alpha)\operatorname{Tr}\left[e^{i\pi\hat{N}}\hat{D}(\eta-\alpha)\right]D(\alpha|\eta/2) \\
&= \int d\bar{\alpha}d\alpha f(\bar{\alpha},\alpha)\prod_k\left[(\alpha_k-\eta_k)(\bar{\alpha}_k-\bar{\eta}_k)\right]D(\alpha|\eta/2) \\
&= f(\bar{\eta},\eta),
\end{aligned}
$$

where $D(\xi|\eta) \equiv e^{\bar{\xi}\eta-\bar{\eta}\xi}$ is the Grassmann analogy of the usual Fourier transformation kernel for complex variables. We should note that the parity of the operators in $\mathcal{K}$ and the functions in $\mathcal{G}$ has significance in making these mappings. See Appendix.A for some details and useful formulas.

These two mappings $\Theta$ and $\Omega$ between $\mathcal{K}$ and $\mathcal{G}$ form the foundation of the characteristic approach. Obviously these mappings have nothing to do with the special form of the Hamiltonian and the dissipators. They are general and only depend on the degree of freedom. For example, for a system with $N$ degree of freedom, we have

$$
\Theta(\hat{c}_i^\dagger) = -\bar{\xi}_i\prod_{k\neq i}\xi_k\bar{\xi}_k\,, \quad \Theta(\hat{c}_i) = \xi_i\prod_{k\neq i}\xi_k\bar{\xi}_k\,, \quad \Theta(\hat{c}_i^\dagger\hat{c}_i) = 2^{N-1}e^{\bar{\xi}_i\xi_i/2}\,.
$$

Some more useful mappings are given in Appendix.A. We stress that although in the following sections we would discuss a special model which can be solved exactly, this does not mean that the *characteristic function approach* is only applicable to such special models.

Using these mappings we can transform problems in the Liouville-Fock space, for example, the quantum master equation, to problems in the Grassmann algebra, and transform back if necessary. The advantage is that for functions in the Grassmann algebra we have rich analytic and algebraic tools [80]. For example, the trace in the Fock space can be transformed to an integration over the Grassmann variables, while the average of one-body or two-body observables with respect to any density matrix $\rho$ can be transformed to partial derivatives of the corresponding characteristic function [see Eq.(13) for an example]. Furthermore, due to the similarity between our method and the phase-space approach in quantum optics [76, 77], we can also borrow concepts and techniques used for bosons. For example, we can define phase-space distribution functions such as the Husimi-Kano $Q$-function or Glauber-Sudarshan $P$-function for fermions. More systematic developments of the formalism long this line deserve further investigations. See Appendix.A for a simple example for the $Q$-function.

Now consider an open system of $N$ sites with spinless fermions, whose dynamics is described by the Gorini-Kossakorsky-Sudarshan-Lindblad (GKSL) equation [36, 37] with Liouvillian $\mathcal{L}$ (we set $\hbar = 1$)

$$
\partial_t\rho = \mathcal{L}(\rho) = -i[\hat{H},\rho] + \sum_\mu\left(2\hat{L}_\mu\rho\hat{L}_\mu^\dagger - \{\hat{L}_\mu^\dagger\hat{L}_\mu,\rho\}\right), \tag{3}
$$

where $\hat{L}_\mu$ are the so-called Lindblad or jump operators. Although the *characteristic function approach* is a quite general theory for treating open fermion systems, here, for simplicity and as a starting point, we focus on general quadratic Hamiltonians

$$
\hat{H} = \frac{1}{2}(\hat{c}^\dagger, \hat{c})\mathbb{H}\begin{pmatrix}\hat{c}\\\hat{c}^\dagger\end{pmatrix}, \tag{4}
$$

and linear Lindbaldian operators

$$\hat{L}_\mu = L_\mu^\dagger \begin{pmatrix} \hat{c} \\ \hat{c}^\dagger \end{pmatrix}, \quad \hat{L}_\mu^\dagger = (\hat{c}^\dagger, \hat{c}) L_\mu, \tag{5}$$

where $(\hat{c}^\dagger, \hat{c}) = (\hat{c}_1^\dagger, \hat{c}_2^\dagger, \ldots, \hat{c}_N^\dagger, \hat{c}_1, \ldots, \hat{c}_N)$, $L_\mu(L_\mu^\dagger)$ are $2N$-dimensional column (row) vectors, while $\mathbb{H}$ is a $2N \times 2N$ matrix satisfying the symmetry requirement

$$\mathbb{H} + \tau_x \mathbb{H}^T \tau_x = 0, \tag{6}$$

where $\tau_{x,y,z}$ denote the Pauli matrices in the particle-hole subspace. Although such a *quadratic Lindbaldian* can be solved exactly by various methods [55–63], the computation of dynamical correlations of nonlocal operators is still a challenging problem. In the characteristic function approach we transform the quantum master equation of the density matrix into an equation for its characteristic function $F(\bar{\xi}, \xi) \equiv \text{Tr}[\hat{D}(\xi)\rho]$,

$$\partial_t F + (\bar{\xi}, \xi)[i\mathbb{H} + \mathbb{X}_+] \begin{pmatrix} \bar{\partial} \\ \partial \end{pmatrix} F = -\frac{1}{2}(\bar{\xi}, \xi)\mathbb{X}_- \begin{pmatrix} \xi \\ \bar{\xi} \end{pmatrix} F, \tag{7}$$

where

$$\mathbb{X}_\pm = \sum_\mu \left[ L_\mu L_\mu^\dagger \pm \tau_x (L_\mu L_\mu^\dagger)^* \tau_x \right], \tag{8}$$

and $(\bar{\partial}, \partial) = (\partial/\partial\bar{\xi}_1, \ldots, \partial/\partial\bar{\xi}_N, \partial/\partial\xi_1, \ldots, \partial/\partial\xi_N)$. See Appendix.B for the details of the derivation. We comment that for a general Liouvillian the equation for $F(\bar{\xi}, \xi)$ would include higher derivatives with respect to $\bar{\xi}, \xi$ and hence can seldom be solved exactly. Fortunately, for the quadratic Hamiltonian [Eq.(4)] and linear dissipators [Eq.(5)] the equation (7) is a first order partial differential equation which an be solved exactly by standard technique. The solution with an arbitrary initial condition $F(\bar{\xi}, \xi; t = 0) = F_0(\bar{\xi}, \xi)$ is

$$F = F_0 \left[ (\bar{\xi}, \xi)\mathbb{Q}(t) \right] \exp\left[ -\frac{1}{2}(\bar{\xi}, \xi)\mathbb{M}(t) \begin{pmatrix} \xi \\ \bar{\xi} \end{pmatrix} \right], \tag{9}$$

where the arguments of $F(\bar{\xi}, \xi; t)$ have not been written explicitly for brevity, and

$$\mathbb{Q}(t) = e^{-(\mathbb{X}_+ + i\mathbb{H})t}, \quad \bar{\mathbb{Q}}(t) = e^{-(\mathbb{X}_+ - i\mathbb{H})t}, \quad \mathbb{M}(t) = \int_0^t dt' \, \mathbb{Q}(t') \, \mathbb{X}_- \, \bar{\mathbb{Q}}(t'). \tag{10}$$

The solution of Eq.(9) is a linear mapping from $F_0(\bar{\xi}, \xi)$ to $F(\bar{\xi}, \xi; t)$, which will be denoted as $F(\bar{\xi}, \xi; t) = \mathcal{U}_t[F_0(\bar{\xi}, \xi)]$. Obviously, $F(\bar{\xi}, \xi; t) = \Theta[\rho(t)] = \Theta[e^{\mathcal{L}t}(\rho_0)] = \mathcal{U}_t[\Theta(\rho_0)]$, or more generally,

$$\Theta \star e^{\mathcal{L}t} = \mathcal{U}_t \star \Theta, \tag{11}$$

where $\star$ denotes the composition of two linear mappings. We comment that the structure of the solution Eq.(9) is very similar to its bosonic counterpart (see, for example, the work by T. Heinosaari *et al.* [81]).

Furthermore, we argue that the $2^{2N}$ eigenvalues of the Liouvillian $\mathcal{L}$ can be constructed from the eigenvalues $\lambda_k$ of $\mathbb{X}_+ + i\mathbb{H}$ as $-\sum_k \nu_k \lambda_k$, where $\nu_k \in \{0, 1\}$. This is quite similar to the expression of the Liouvillian spectrum in terms of the so-called "rapidities" in the third quantization method [54]. To show this, let's suppose that $\{\lambda_k\}$ are the eigenvalues and $\{|\varphi_k^{R(L)}\rangle\}$ the right (left) eigenvectors of $\mathbb{X}_+ + i\mathbb{H}$. Then

$$\mathbb{Q}(t) = \sum_{k=1}^{2N} e^{-\lambda_k t} |\varphi_k^R\rangle\langle\varphi_k^L|, \quad \bar{\mathbb{Q}}(t) = \tau_x [\mathbb{Q}(t)]^T \tau_x = \sum_{k=1}^{2N} e^{-\lambda_k t} \, \tau_x |\varphi_k^{L*}\rangle\langle\varphi_k^{R*}| \tau_x.$$

From Eq.(9) we know that the characteristic function can be expanded as

$$F(t) = \sum_{\{\nu_k\}} F_{\{\nu_k\}} e^{-t \sum_k \nu_k \lambda_k}.$$

This is because the time dependence of $F(t)$ is completely encoded in $\mathbb{Q}(t)$ and $\bar{\mathbb{Q}}(t)$, which can be expanded in terms of their corresponding eigenvectors. Therefore, by mapping from $\mathcal{G}$ to $\mathcal{K}$, the density matrix can also be expanded as

$$\rho(t) = \sum_{\{\nu_k\}} \rho_{\{\nu_k\}} e^{-t \sum_k \nu_k \lambda_k},$$

from which we can deduce the spectrum of the Liouvillian $\mathcal{L}$. As a result, the Liouvillian gap is given by the minimum value of $\mathrm{Re}(\lambda_k)$.

Now let's compare the characteristic function approach with other methods, especially with the "third quantization method" [54–57]. (i) One straightforward way to compute the dynamical correlations is to use the equations of motion method, which depends on commutations between the observables and the Hamiltonian/dissipators. For one-body or two-body observables, such commutations can give a set of closed equations that can be easily solved. However, this is impractical for nonlocal operators since the commutations would induce more and more complicated operators and the resulting set of equations is very large. (ii) The third quantization method defines $4N$ linear maps over the Liouville-Fock space $\mathcal{K}$ which satisfy canonical anticommutation relations. The key quantity is a $4N \times 4N$ matrix whose eigenvalues are paired as $\beta_j, -\beta_j, j = 1, 2, \ldots, 2N$, with $\mathrm{Re}\beta_j \geq 0$. In contrast, the key matrix in the characteristic function approach is $\mathbb{X}_+ + i\mathbb{H}$, which has dimension $2N \times 2N$. (iii) In third quantization method, the steady state is implicitly defined as the right vacuum of the Liouvillian, while in our method the steady state can be given explicitly [see Eqs.(12) and (59)]. (iv) For higher-order observables, the third quantization method relies on the Wick's theorem, which is impractical for computing correlations of nonlocal operators. In contrast our method presents a practical way. (v) Of course, the characteristic function approach has its own disadvantages. For example, the $\Omega$ and $\Theta$ mappings may be difficult to do for some complicated operators and functions. In addition, the anticommutation nature of the Grassmann variables asks for meticulous care in calculations. A researcher who is not familiar with the Grassmann algebra may make mistakes unknowingly.

## 2.2 Physical observables

Now let's discuss some physical properties of the open fermion system based on the solution given by Eq.(9). We remark that the results in this subsection could also be obtained by other methods [54–63], however, here we briefly present these results to show the completeness of our new method.

(i) The steady state can be obtained by taking the limit $t \to \infty$. If all the eigenvalues $\lambda_\alpha$ of $(\mathbb{X}_+ + i\mathbb{H})$ have positive real parts, i.e., $\mathrm{Re}\lambda_\alpha > 0$, then $\mathbb{Q}(t) \to 0$ while $\mathbb{M}(t) \to \mathbb{M}_\infty$ as $t \to \infty$, and the characteristic function approaches to

$$F_\infty = \exp\left[ -\frac{1}{2}(\bar{\xi}, \xi)\mathbb{M}_\infty \begin{pmatrix} \xi \\ \bar{\xi} \end{pmatrix} \right]. \tag{12}$$

This is a Gaussian state determined solely by the Hamiltonian and the dissipators, independent of the initial state. On the contrary, if some eigenvalues $\lambda_\alpha$ have zero real parts, $\mathbb{Q}(t)$ may not approach to zero and the system would have no unique steady state.

(ii) The covariance (or equal-time correlation) matrix can be expressed in terms of the characteristic function:

$$\mathbb{C} \equiv \left\langle \begin{pmatrix} \hat{c} \\ \hat{c}^{\dagger} \end{pmatrix} (\hat{c}^{\dagger}, \hat{c}) \right\rangle = \frac{1}{2}\mathbb{1} + \begin{pmatrix} \bar{\partial} \\ \partial \end{pmatrix} (\partial, \bar{\partial}) F(\bar{\xi}, \xi) \bigg|_{0}, \tag{13}$$

where $f(\bar{\xi}, \xi)|_0$ means taking $\xi = \bar{\xi} = 0$ at last. From the equation for $F(\bar{\xi}, \xi)$ we can deduce the equation of motion for this covariance matrix:

$$\partial_t \mathbb{C} = [\mathbb{C}, i\mathbb{H}] - \{\mathbb{C}, \mathbb{X}_+\} + (\mathbb{X}_+ + \mathbb{X}_-),$$

where $\{\cdot, \cdot\}$ denotes anticommutation relation. For the steady state described by Eq.(12), we have

$$\mathbb{C}_\infty = \frac{1}{2}\left(\mathbb{1} + \mathbb{M}_\infty - \tau_x \mathbb{M}_\infty^T \tau_x\right) = \frac{1}{2}\mathbb{1} + \mathbb{M}_\infty. \tag{14}$$

(iii) The nonequilibrium Green's functions, which describe the excitations in the steady state, can also be expressed in terms of the characteristic function. For example, the retarded Green function can be obtained through

$$G^{\mathrm{R}}(t) \equiv -i\theta(t) \left\langle \left\{ \begin{pmatrix} \hat{c}(t) \\ \hat{c}^{\dagger}(t) \end{pmatrix}, (\hat{c}^{\dagger}, \hat{c}) \right\} \right\rangle_s = -i\theta(t) \begin{pmatrix} \bar{\partial} \\ \partial \end{pmatrix} \mathcal{U}_t \left[ (\bar{\xi}, \xi) F_s(\bar{\xi}, \xi) \right] \bigg|_0, \tag{15}$$

where $F_s$ is the characteristic function of the steady state $\rho_s$. For the Gaussian state given by Eq.(12) the retarded Green function simply reads $G^{\mathrm{R}}(t) = -i\theta(t)\mathbb{Q}(t)$.

(iv) Furthermore, the dynamical response function or the density-density correlation function can be defined as

$$D_{ij}(t) \equiv -i\theta(t)\langle[\hat{n}_i(t), \hat{n}_j]\rangle, \tag{16}$$

where $\hat{n}_j = \hat{c}_j^{\dagger}\hat{c}_j$. Using the same technique as that for the Green's functions we can obtain its expression in the steady state given by Eq.(12):

$$\begin{aligned} D_{ij}(t) &= -i\theta(t)\left\{ [\mathbb{Q}\mathbb{M}_\infty]_{ij}[\bar{\mathbb{Q}}]_{ji} - [\mathbb{Q}]_{ij}[\mathbb{M}_\infty\bar{\mathbb{Q}}]_{ji} \right. \\ &\quad \left. - [\mathbb{Q}\mathbb{M}_\infty]_{i+N,j}[\bar{\mathbb{Q}}]_{j,i+N} + [\mathbb{Q}]_{i+N,j}[\mathbb{M}_\infty\bar{\mathbb{Q}}]_{j,i+N} \right\}, \end{aligned} \tag{17}$$

where the time dependence of $\mathbb{Q}(t)$ and $\bar{\mathbb{Q}}(t)$ have not been written explicitly for brevity. In the same manner all dynamical correlation functions of local operators can be obtained by taking derivatives of the characteristic function, just as in Eq.(15).

## 2.3 Dynamical correlations of nonlocal operators

Now we turn to our main problem: the dynamical correlations of nonlocal operators. We would call the exponential of a general bilinear form of fermion creation and annihilation operators as *Gaussian operators*, and denote them as

$$\hat{\Gamma}_2(\mathbb{K}) \equiv \exp\left[\frac{1}{2}(\hat{c}^{\dagger}, \hat{c})\mathbb{K}\begin{pmatrix} \hat{c} \\ \hat{c}^{\dagger} \end{pmatrix}\right], \tag{18}$$

where $\mathbb{K}$ is a $2N \times 2N$ matrix satisfying $\mathbb{K} + \tau_x \mathbb{K}^T \tau_x = 0$. String operators can be treated as a special kind of Gaussian operators. We comment that the requirement of $\mathbb{K}$ is not necessary but it would make the following formulas more concise. First, since $\hat{c}_i^{\dagger}\hat{c}_j$ and $\hat{c}_j\hat{c}_i^{\dagger}$ are not independent, the matrix $\mathbb{K}$ can be written in many different forms up to an overall multiplier of the Gaussian operator. The above requirement may remove this ambiguity by taking one special choice. Second, this special choice is very convenient in making the computations in

the characteristic function approach. For example, in the $\Theta$ mappings given by Eqs.(48) and (49) we require the matrix $\mathbb{K}$ to satisfy the above requirement, otherwise the equation would be lengthy.

According to the quantum regression formula [76], two-time correlations of $\hat{O}_1(t), t \geq 0$, and $\hat{O}_2(0)$ with respect to a density matrix $\rho(0)$ are given by

$$\langle \hat{O}_1(t)\hat{O}_2(0)\rangle = \text{Tr}\left\{\hat{O}_1(0)e^{\mathcal{L}t}\left[\hat{O}_2(0)\rho(0)\right]\right\},$$
$$\langle \hat{O}_2(0)\hat{O}_1(t)\rangle = \text{Tr}\left\{\hat{O}_1(0)e^{\mathcal{L}t}\left[\rho(0)\hat{O}_2(0)\right]\right\}.$$

Considering Gaussian states and Gaussian operators, the above correlations would have the same form up to a $c$-number factor,

$$\text{Type-I:} \qquad \text{Tr}\left\{\hat{\Gamma}_2(\mathbb{K}_1)e^{\mathcal{L}t}\left[\hat{\Gamma}_2(\mathbb{K}_2)\hat{\Gamma}_2(\mathbb{K}_0)\right]\right\}. \tag{19}$$

In addition, we are also interested in single-particle correlations such as the Green's functions. Here we consider more generally the dynamical correlations of nonlocal single-particle operators, i.e., the single-particle creation/annhilation operators multiplied by a string or Gaussian operator. However, in fermionic systems we should note that the standard version of the quantum regression formula [76], which assumes $\hat{O}_{1,2}$ to be bosonic, does not apply due to the fact that the single-particle operators contain an odd number of fermionic operators. For a proof from the first principle please refer to the work by F. Schwarz *et al.* [82]. The appropriate Liouvillian reads

$$\mathcal{L}_f(\circ) = -i[\hat{H}, \circ] + \sum_{\mu}\left(-2\hat{L}_\mu \circ \hat{L}_\mu^\dagger - \{\hat{L}_\mu^\dagger \hat{L}_\mu, \circ\}\right).$$

The relation between $\mathcal{L}$ and $\mathcal{L}_f$ is discussed in Appendix.C. Then the dynamical correlations of nonlocal single-particle operators in a Gaussian state take the general form

$$\text{Type-II:} \qquad \text{Tr}\left\{\begin{pmatrix}\hat{c}\\\hat{c}^\dagger\end{pmatrix}\hat{\Gamma}_2(\mathbb{K}_1)e^{\mathcal{L}_f t}\left[\hat{\Gamma}_2(\mathbb{K}_2)(\hat{c}^\dagger, \hat{c})\hat{\Gamma}_2(\mathbb{K}_0)\right]\right\}, \tag{20}$$

where the trace is take over the Fock space and hence the result is a $2N \times 2N$ matrix.

We will give explicit formulas for these correlation functions. Before that, it's convenient to define the following matrices: $\mathbb{B}_0 \equiv \left[\mathbb{1} + e^{\mathbb{K}_0}\right]^{-1}$, $\mathbb{W}_{20} \equiv e^{\mathbb{K}_2}e^{\mathbb{K}_0}$, $\mathbb{W}_{02} \equiv e^{\mathbb{K}_0}e^{\mathbb{K}_2}$,

$$\mathbb{B}_{20} \equiv \frac{1}{2}\mathbb{1} + \frac{1}{2}\mathbb{Q}(t)\frac{\mathbb{1} - \mathbb{W}_{20}}{\mathbb{1} + \mathbb{W}_{20}}\bar{\mathbb{Q}}(t) + \mathbb{M}(t),$$

$$\mathbb{B}_{02} \equiv \frac{1}{2}\mathbb{1} + \frac{1}{2}\mathbb{Q}(t)\frac{\mathbb{1} - \mathbb{W}_{02}}{\mathbb{1} + \mathbb{W}_{02}}\bar{\mathbb{Q}}(t) + \mathbb{M}(t),$$

and $\mathbb{R}_{20} \equiv \mathbb{B}_0 + e^{\mathbb{K}_2}(\mathbb{1} - \mathbb{B}_0)$, $\mathbb{R}_{02} \equiv \mathbb{B}_0 + (\mathbb{1} - \mathbb{B}_0)e^{\mathbb{K}_2}$, $\mathbb{S}_{20} \equiv \mathbb{B}_{20} + (\mathbb{1} - \mathbb{B}_{20})e^{\mathbb{K}_1}$, $\mathbb{S}_{02} \equiv \mathbb{B}_{02} + (\mathbb{1} - \mathbb{B}_{02})e^{\mathbb{K}_1}$.

Using the three linear mappings $\Omega, \Theta$ and $\mathcal{U}_t$, we have

$$\text{Tr}\left\{\hat{\Gamma}_2(\mathbb{K}_1)e^{\mathcal{L}t}\left[\hat{\Gamma}_2(\mathbb{K}_2)\hat{\Gamma}_2(\mathbb{K}_0)\right]\right\} = \text{Tr}\left\{\hat{\Gamma}_2(\mathbb{K}_1)\Omega \star \mathcal{U}_t \star \Theta\left[\hat{\Gamma}_2(\mathbb{K}_2)\hat{\Gamma}_2(\mathbb{K}_0)\right]\right\}.$$

Now we compute the three mappings one by one:

(i). $\qquad \Theta\left[\hat{\Gamma}_2(\mathbb{K}_2)\hat{\Gamma}_2(\mathbb{K}_0)\right] = \sqrt{\det(\mathbb{1} + \mathbb{W}_{20})}\exp\left[-\frac{1}{2}(\bar{\xi}, \xi)\frac{1}{\mathbb{1} + \mathbb{W}_{20}}\begin{pmatrix}\xi\\\bar{\xi}\end{pmatrix}\right],$

(ii). $\qquad \mathcal{U}_t \star \Theta\left[\hat{\Gamma}_2(\mathbb{K}_2)\hat{\Gamma}_2(\mathbb{K}_0)\right]$

$\qquad = \sqrt{\det(\mathbb{1} + \mathbb{W}_{20})}\exp\left[-\frac{1}{2}(\bar{\xi}, \xi)\left(\mathbb{Q}(t)\frac{1}{\mathbb{1} + \mathbb{W}_{20}}\bar{\mathbb{Q}}(t) + \mathbb{M}(t)\right)\begin{pmatrix}\xi\\\bar{\xi}\end{pmatrix}\right]$

$\qquad = \sqrt{\det(\mathbb{1} + \mathbb{W}_2)}\exp\left[-\frac{1}{2}(\bar{\xi}, \xi)\mathbb{B}_{20}\begin{pmatrix}\xi\\\bar{\xi}\end{pmatrix}\right],$

(iii). $\qquad \Omega \star \mathcal{U}_t \star \Theta\left[\hat{\Gamma}_2(\mathbb{K}_2)\hat{\Gamma}_2(\mathbb{K}_0)\right] = \sqrt{\det(\mathbb{1} + \mathbb{W}_{20})}\sqrt{\det\mathbb{B}_{20}}\,\hat{\Gamma}_2(\mathbb{K}_{B_{20}}),$

where $\mathbb{K}_{B_{20}}$ is defined through $\mathbb{B}_{20}(\mathbb{1} + e^{\mathbb{K}_{B_{20}}}) = \mathbb{1}$. Note that in (ii) we have changed the matrix in the exponential to $\mathbb{B}_{20}$ to satisfy the requirement $\mathbb{B}_{20} + \tau_x \mathbb{B}_{20}^T \tau_x = \mathbb{1}$. Finally, taking the trace gives the result:

$$\text{Tr}\left\{\hat{\Gamma}_2(\mathbb{K}_1)e^{\mathcal{L}t}\left[\hat{\Gamma}_2(\mathbb{K}_2)\hat{\Gamma}_2(\mathbb{K}_0)\right]\right\} = \sqrt{\det(\mathbb{1} + \mathbb{W}_{20})\det\mathbb{S}_{20}}. \tag{21}$$

When $t = 0$, $\mathbb{Q} = \mathbb{1}$, $\mathbb{M} = 0$, and $\mathbb{B}_{20} = [1 + \mathbb{W}_{20}]^{-1}$, then we can obtain the static correlation function $\text{Tr}\left\{\hat{\Gamma}_2(\mathbb{K}_1)\hat{\Gamma}_2(\mathbb{K}_2)\hat{\Gamma}_2(\mathbb{K}_0)\right\} = \sqrt{\det[\mathbb{1} + e^{\mathbb{K}_1}e^{\mathbb{K}_2}e^{\mathbb{K}_0}]}$.

Two remarks should be added here. (1) An issue of the determinant formulas is that the sign of the square root of the determinant has to be determined. In some simple cases the square root of a determinant can be rewritten as a Pfaffian [83]. However, this is difficult for general cases, especially for products of several Gaussian operators. In practical calculations the sign can be determined as follows. For $Z(\mathbb{A}) = \sqrt{\det[\mathbb{1} + e^{\mathbb{A}}]}$, we consider $Z(\lambda\mathbb{A})$, which should be an analytic function of $\lambda$. This determines the correct way of taking the sign of the square root: the sign has to be taken so that $Z(\lambda\mathbb{A})$ is everywhere analytic and at $\lambda = 0$ one has $Z(0) = 2^N$. (2) Some matrices used in these formulas should satisfy certain symmetry requirements, namely, $\mathbb{A} + \tau_x \mathbb{A}^T \tau_x = 0$ for $\mathbb{A} = \mathbb{H}, \mathbb{M}(t), \mathbb{K}_{0,1,2}$, while $\mathbb{A} + \tau_x \mathbb{A}^T \tau_x = \mathbb{1}$ for $\mathbb{A} = \mathbb{B}_0, \mathbb{B}_{20}$ and $\mathbb{B}_{02}$.

Now consider the dynamical correlations of nonlocal single-particle operators, which takes the type-II form of Eq.(20). Even for quadratic Lindbladian these correlations are difficult to compute. Here we use the characteristic function approach to solve this problem. The correlation can be rewritten as

$$\text{Tr}\left\{\begin{pmatrix} \hat{c} \\ \hat{c}^\dagger \end{pmatrix}\hat{\Gamma}_2(\mathbb{K}_1)e^{\mathcal{L}_f t}\left[\hat{\Gamma}_2(\mathbb{K}_2)(\hat{c}^\dagger, \hat{c})\hat{\Gamma}_2(\mathbb{K}_0)\right]\right\}$$

$$= \text{Tr}\left\{\begin{pmatrix} \hat{c} \\ \hat{c}^\dagger \end{pmatrix}\hat{\Gamma}_2(\mathbb{K}_1)e^{i\pi\hat{N}}e^{\mathcal{L}t}\left[e^{i\pi\hat{N}}\hat{\Gamma}_2(\mathbb{K}_2)(\hat{c}^\dagger, \hat{c})\hat{\Gamma}_2(\mathbb{K}_0)\right]\right\}$$

$$= \text{Tr}\left\{\begin{pmatrix} \hat{c} \\ \hat{c}^\dagger \end{pmatrix}\hat{\Gamma}_2(\mathbb{K}_1)e^{i\pi\hat{N}}\Omega \star \mathcal{U}_t \star \Theta\left[e^{i\pi\hat{N}}\hat{\Gamma}_2(\mathbb{K}_2)(\hat{c}^\dagger, \hat{c})\hat{\Gamma}_2(\mathbb{K}_0)\right]\right\}.$$

Then we can do the three mappings $\Omega, \mathcal{U}_t$ and $\Theta$ one by one, and make the trace to obtain the final result:

$$\text{Tr}\left\{\begin{pmatrix} \hat{c} \\ \hat{c}^\dagger \end{pmatrix}\hat{\Gamma}_2(\mathbb{K}_1)e^{\mathcal{L}_f t}\left[\hat{\Gamma}_2(\mathbb{K}_2)(\hat{c}^\dagger, \hat{c})\hat{\Gamma}_2(\mathbb{K}_0)\right]\right\}$$

$$= \frac{\sqrt{\det[\mathbb{R}_{20}]\det[\mathbb{S}_{20}]}}{\sqrt{\det[\mathbb{B}_0]}}e^{\mathbb{K}_1}[\mathbb{S}_{20}]^{-1}\mathbb{Q}(t)\mathbb{B}_0[\mathbb{R}_{20}]^{-1}e^{\mathbb{K}_2}. \tag{22}$$

By exchanging $\mathbb{K}_2$ and $\mathbb{K}_0$, we have another form

$$\text{Tr}\left\{\begin{pmatrix} \hat{c} \\ \hat{c}^\dagger \end{pmatrix}\hat{\Gamma}_2(\mathbb{K}_1)e^{\mathcal{L}_f t}\left[\hat{\Gamma}_2(\mathbb{K}_0)(\hat{c}^\dagger, \hat{c})\hat{\Gamma}_2(\mathbb{K}_2)\right]\right\}$$

$$= \frac{\sqrt{\det[\mathbb{R}_{02}]\det[\mathbb{S}_{02}]}}{\sqrt{\det[\mathbb{B}_0]}}e^{\mathbb{K}_1}[\mathbb{S}_{02}]^{-1}\mathbb{Q}(t)[\mathbb{R}_{02}]^{-1}(\mathbb{1} - \mathbb{B}_0). \tag{23}$$

We would not give the technical details here since the procedure is lengthy but straightforward. We just give three remarks.

(i) If $\mathbb{K}_1 = \mathbb{K}_2 = 0$, then $\mathbb{R}_{20} = \mathbb{S}_{20} = \mathbb{1}$, and the correlations would reduce to that of local operators:

$$\text{Tr}\left\{\begin{pmatrix} \hat{c} \\ \hat{c}^\dagger \end{pmatrix}e^{\mathcal{L}_f t}\left[(\hat{c}^\dagger, \hat{c})\hat{\Gamma}_2(\mathbb{K}_0)\right]\right\} = \mathbb{Q}(t)\frac{\sqrt{\det[\mathbb{1} + e^{\mathbb{K}_0}]}}{\mathbb{1} + e^{\mathbb{K}_0}}.$$

(ii)If $t = 0$, then $\mathbb{Q} = \mathbb{1}, \mathbb{M} = 0$ and $\mathbb{B}_{20} = (\mathbb{1} + \mathbb{W}_{20})^{-1}$, and the result would reduce to the static correlations:

$$\text{Tr}\left\{\begin{pmatrix} \hat{c} \\ \hat{c}^\dagger \end{pmatrix}\hat{\Gamma}_2(\mathbb{K}_1)\hat{\Gamma}_2(\mathbb{K}_2)(\hat{c}^\dagger, \hat{c})\hat{\Gamma}_2(\mathbb{K}_0)\right\} = \frac{\sqrt{\det\left[\mathbb{1} + e^{\mathbb{K}_1}e^{\mathbb{K}_2}e^{\mathbb{K}_0}\right]}}{\mathbb{1} + e^{\mathbb{K}_1}e^{\mathbb{K}_2}e^{\mathbb{K}_0}}e^{\mathbb{K}_1}e^{\mathbb{K}_2}, \qquad (24)$$

(iii) If we consider the correlations in the steady state given by Eq.(12), we should note that the corresponding density matrix is

$$\rho_s = \sqrt{\det\left(\frac{1}{2}\mathbb{1} + \mathbb{M}_\infty\right)}\hat{\Gamma}_2(\mathbb{K}_0), \qquad (25)$$

where $\mathbb{K}_0$ is determined by $\left(\frac{1}{2}\mathbb{1} + \mathbb{M}_\infty\right)\left(\mathbb{1} + e^{\mathbb{K}_0}\right) = \mathbb{1}$, and the corresponding $\mathbb{B}_0 = \frac{1}{2}\mathbb{1} + \mathbb{M}_\infty$.

# 3 Kitaev chain with boundary dissipation

In this section we take the Kitaev chain [84] with boundary dissipation as an example to illustrate the general techniques developed above.

## 3.1 The Model and the spectrum

The Hamiltonian is

$$\hat{H}_K = \sum_{l=1}^{N-1}\left[(J\hat{c}_l^\dagger\hat{c}_{l+1} + \Delta\hat{c}_l\hat{c}_{l+1}) + \text{h.c.}\right] - \mu\sum_{l=1}^{N}\hat{c}_l^\dagger\hat{c}_l, \qquad (26)$$

which can be rewritten as a bilinear form of Eq.(4). We consider single-particle gain and loss dissipators,

$$\hat{L}_{j+} = \sqrt{\gamma_{j+}}\,\hat{c}_j^\dagger, \quad \hat{L}_{j-} = \sqrt{\gamma_{j-}}\,\hat{c}_j. \qquad (27)$$

For simplicity of this illustrating example we take dissipations which act only on the first and last sites, i.e., $\gamma_{1\pm} = \gamma_{N\pm} = \gamma_\pm$ and all other dissipators vanish. With this setting the model is essentially equivalent to the boundary-driven XY spin chain [54–57, 85]. Therefore we can immediately infer that there is an NQPT [54] in the $\Delta$-$\mu$ space at the critical lines $\pm\mu_c/J = \pm 2[1 - (\Delta/J)^2]$. Namely, there is the so called long-range magnetic correlation (LRMC) phase for $|\mu| < \mu_c$ and the non-LRMC phase for $|\mu| > \mu_c$. We remark that the symmetric dissipative driving on the two ends of the chain is not necessary here. We choose this special setting just for simplicity and to show that the nonlocal excitations can exhibit asymmetric spatial propagation even for symmetric Hamiltonian and dissipations [see Fig.2 in the following]. If the driving is not symmetric, the NQPT still exists and most of the following results hold qualitatively, except for the result about the spatial symmetry of the local Green's function [as shown in Fig.2]. Notably, it has been found that boundary dephasing on a single boundary could enhance the correlation time of the local degree of freedom at the opposite boundary [86]. Similar effect can also exist for linear dissipators at a single edge. However, we would restrict ourselves to the symmetric boundary driving in the following to illustrate the general technique developed above.

As seen from the solution of the quadratic Lindbladian, the dynamics is completely determined by three matrices: $\mathbb{H}$ and $\mathbb{X}_\pm$. In fact, the matrix $\mathbb{X}_+ + i\mathbb{H}$ determines the dissipative dynamics and the Liouvillian spectrum. In Fig.1 we plot the imaginary and real parts of the eigenvalues $\lambda_\alpha, \alpha = 1, 2, \ldots, 2N$ of the matrix $\mathbb{X}_+ + i\mathbb{H}$. The Liouvillian gap can be derived from the smallest value of $\text{Re}(\lambda)$, which approaches to zero and hence signaling an NQPT at

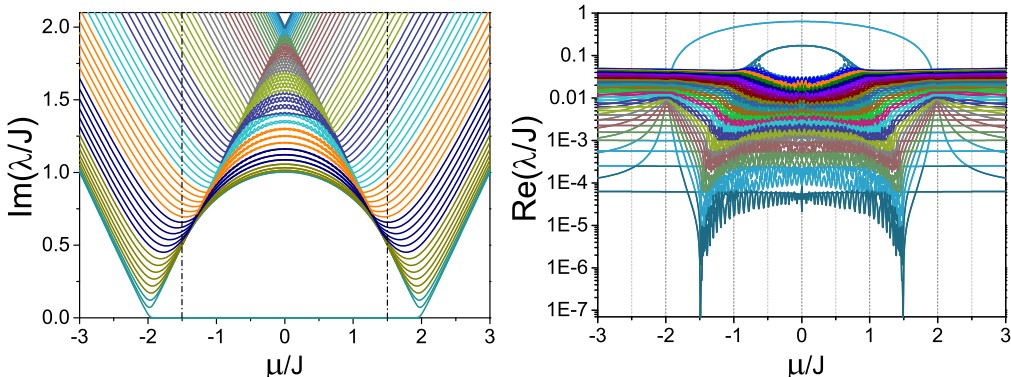

Figure 1: The imaginary and real part of the eigenvalues $\lambda_\alpha$ of $\mathbb{X}_+ + i\mathbb{H}$. Since the imaginary part is symmetric about the origin, only the positive half has been shown. The parameters are chosen as: $\Delta/J = 0.5, \gamma_-/J = 0.5, \gamma_+/J = 0.2$ and $N = 64$. The dashed lines in the left plot denote the critical chemical potential $\pm\mu_c/J = \pm2[1-(\Delta/J)^2] = \pm1.5$. Between the two dashed lines there is a region where the energy levels have may crossings. In the right plot the highest line between $\mu/J = \pm2$ corresponds to the edge modes with $\text{Im}(\lambda/J) = 0$.

$\mu/J = \pm1.5$. Furthermore, two other features can be observed: (i) There are two degenerate modes with $\text{Im}(\lambda) = 0$ when $|\mu/J| \leq 2$. The corresponding left and right eigenvectors are localized at the edges, similar to the Majorana zero modes in the closed system. However, in the steady state phase diagram there is no corresponding topological phase transition at $\mu/J = \pm2$. This is because these edge modes do not contribute to the steady state as a result of the particle-hole symmetry of the edge modes and the matrix $\mathbb{X}_-$. Furthermore, the real part of the eigenvalues of the edge modes has relatively large positive value, so that the edge modes decay very rapidly in the dissipative dynamics.

(ii) In the left plot of Fig.1 we also observe that there is a region where the energy levels have many crossings. This abrupt change of level degeneracy is a characteristic signature of the so-called ESQPT [87]. In fact the level structure is similar to (but different from) that of the nonlinear Kerr oscillator where the ESQPT has been investigated systematically in a recent paper [88]. In the thermodynamic limit $N \to \infty$ the bulk spectrum is insensitive to the boundary dissipation and is given by the spectrum of $\mathbb{H}$,

$$\text{Im}(\lambda) = \pm2J\sqrt{\left(\cos q - \frac{\mu}{2J}\right)^2 + \frac{\Delta^2}{J^2}\sin^2 q}\,, \tag{28}$$

with $q \in (-\pi, \pi]$ (see, e.g., [89, 90]). The structure of this dispersion relation qualitatively changes as the chemical potential crosses the critical values, $\pm\mu_c/J = \pm2[1-(\Delta/J)^2]$. These critical values determine phase boundaries of both the ESQPT and the NQPT. This coincidence suggests us a close relationship between ESQPT and NQPT: in the weak dissipation limit ($\gamma_\pm \to 0$) an NQPT would correspond to an ESQPT, but not the ground-state quantum phase transition. This relationship is an interesting issue that deserves further investigations [91].

## 3.2 The Green's function

Now we compute the dynamics of nonlocal excitations, namely, the Green's functions of the hard-core anyons. In one dimension it's well-known that the hard-core anyons satisfy the exchange statistics

$$\hat{f}_l\hat{f}_m^\dagger + e^{-i\phi\,\text{sgn}(l-m)}\hat{f}_m^\dagger\hat{f}_l = \delta_{lm}\,, \qquad \hat{f}_l\hat{f}_m + e^{i\phi\,\text{sgn}(l-m)}\hat{f}_m\hat{f}_l = 0\,, \tag{29}$$

where

$$\mathrm{sgn}(x) = \begin{cases} 1 & \text{if } x > 0, \\ 0 & \text{if } x = 0, \\ -1 & \text{if } x < 0. \end{cases}$$

They can be transformed to spinless fermions multiplied by a string operator,

$$\hat{f}_l^\dagger \equiv \hat{c}_l^\dagger e^{i\phi \sum_{m \leq l} \hat{n}_m}, \quad \hat{f}_l \equiv e^{-i\phi \sum_{m \leq l} \hat{n}_m} \hat{c}_l. \tag{30}$$

Our motivation of studying such excitations is twofold. First, in this fermion model, string order parameters may be useful to characterize topological properties [65–68]. A natural generalization of these order parameters are string operators with arbitrary parameter $\phi \in [0, \pi]$. Second, if the fermionic Hamiltonian is obtained from a hard-core anyon or hard-core boson (Tonks-Girardeau gas or XY spin chain) model, correlations of such nonlocal operators would have physical significance in the original system. For example, the spectral functions of anyonic excitations can be computed from the dynamical correlations, which has already been done in a recent work [92] by the same author for a one-dimensional model without dissipation. Generalizations to dissipative systems can be readily obtained by using the formalisms developed in this section and would be studied systematically in future works.

Here we express the Green's functions explicitly. For that purpose we define the following matrices:

$$\mathbb{R}_\pm^{j0} \equiv \mathbb{B}_0 + e^{\pm i\phi \tau_z \mathbb{D}_j}(\mathbb{1} - \mathbb{B}_0), \quad \mathbb{R}_\pm^{0j} \equiv \mathbb{B}_0 + (\mathbb{1} - \mathbb{B}_0)e^{\pm i\phi \tau_z \mathbb{D}_j},$$

$$\mathbb{B}_\pm^{j0} \equiv \frac{1}{2}\mathbb{1} + \frac{1}{2}\mathbb{Q}(t)\frac{\mathbb{1} - e^{\pm i\phi \tau_z \mathbb{D}_j} e^{\mathbb{K}_0}}{\mathbb{1} + e^{\pm i\phi \tau_z \mathbb{D}_j} e^{\mathbb{K}_0}}\bar{\mathbb{Q}}(t) + \mathbb{M}(t),$$

$$\mathbb{B}_\pm^{0j} \equiv \frac{1}{2}\mathbb{1} + \frac{1}{2}\mathbb{Q}(t)\frac{\mathbb{1} - e^{\mathbb{K}_0} e^{\pm i\phi \tau_z \mathbb{D}_j}}{\mathbb{1} + e^{\mathbb{K}_0} e^{\pm i\phi \tau_z \mathbb{D}_j}}\bar{\mathbb{Q}}(t) + \mathbb{M}(t),$$

$$\mathbb{S}_{ab}^{j0l} \equiv \mathbb{B}_a^{j0} + (\mathbb{1} - \mathbb{B}_a^{j0})e^{bi\phi \tau_z \mathbb{D}_l}, \quad \mathbb{S}_{ab}^{0jl} \equiv \mathbb{B}_a^{0j} + (\mathbb{1} - \mathbb{B}_a^{0j})e^{bi\phi \tau_z \mathbb{D}_l},$$

where $a, b = \pm$, $\mathbb{B}_0 = \frac{1}{2}\mathbb{1} + \mathbb{M}_\infty$, $\tau_z \mathbb{D}_j$ means $\tau_z \otimes \mathbb{D}_j$, and $\mathbb{D}_j$ is a diagonal $N \times N$ matrix with diagonal elements $(\mathbb{D}_j)_{mm} = 1$ if $m \leq j$ and 0 otherwise.

First, the greater Green's function for $t > 0$ reads

$$\begin{aligned} iG_{lj}^>(t) &= \langle \hat{f}_l(t)\hat{f}_j^\dagger \rangle = \mathrm{Tr}\left\{ e^{-i\phi \hat{Q}_l} \hat{c}_l e^{\mathcal{L}_f t}\left[\hat{c}_j^\dagger e^{i\phi \hat{Q}_j}\rho_s\right]\right\} \\ &= e^{i\phi(j-l)/2}\sqrt{\det \mathbb{B}_0}\,\mathrm{Tr}\left\{\hat{c}_l \hat{\Gamma}_2(-i\phi \tau_z \mathbb{D}_l)e^{\mathcal{L}_f t}\left[\hat{\Gamma}_2(i\phi \tau_z \mathbb{D}_j)\hat{c}_j^\dagger \hat{\Gamma}_2(\mathbb{K}_0)\right]\right\}, \end{aligned}$$

where the average $\langle \cdot \rangle$ is taken in the steady state. Using Eq.(22) and setting $\mathbb{K}_1 = -i\phi \tau_z \mathbb{D}_l$, $\mathbb{K}_2 = i\phi \tau_z \mathbb{D}_j$, we obtain

$$iG_{lj}^>(t) = e^{i\phi(j-l)/2}\sqrt{\det \mathbb{R}_+^{j0} \det \mathbb{S}_{+-}^{j0l}}\left\{\left[\mathbb{S}_{+-}^{j0l}\right]^{-1}\mathbb{Q}\mathbb{B}_0\left[\mathbb{R}_+^{j0}\right]^{-1}\right\}_{lj}. \tag{31}$$

Similarly we can obtain

$$iG_{lj}^>(-t) = e^{i\phi(j-l)/2}\sqrt{\det \mathbb{R}_-^{0l} \det \mathbb{S}_{-+}^{0lj}}\left\{\left[\mathbb{S}_{-+}^{0lj}\right]^{-1}\mathbb{Q}\left[\mathbb{R}_-^{0l}\right]^{-1}(\mathbb{1} - \mathbb{B}_0)\right\}_{N+j,N+l}. \tag{32}$$

We can prove that they satisfy the relation, $iG_{jl}^>(-t) = \left[iG_{lj}^>(t)\right]^*$.

Second, the lesser Green's function $iG_{lj}^<(t) = \langle \hat{f}_j^\dagger \hat{f}_l(t) \rangle$ for $t > 0$ can be obtained in a similar manner:

$$iG_{lj}^<(t) = e^{i\phi(j-l)/2}\sqrt{\det \mathbb{R}_+^{0j} \det \mathbb{S}_{+-}^{0jl}}\left\{\left[\mathbb{S}_{+-}^{0jl}\right]^{-1}\mathbb{Q}\left[\mathbb{R}_+^{0j}\right]^{-1}(\mathbb{1} - \mathbb{B}_0)\right\}_{lj}, \tag{33}$$

$$iG_{lj}^<(-t) = e^{i\phi(j-l)/2}\sqrt{\det \mathbb{R}_-^{l0} \det \mathbb{S}_{-+}^{l0j}}\left\{\left[\mathbb{S}_{-+}^{l0j}\right]^{-1}\mathbb{Q}\mathbb{B}_0\left[\mathbb{R}_-^{l0}\right]^{-1}\right\}_{N+j,N+l}. \tag{34}$$

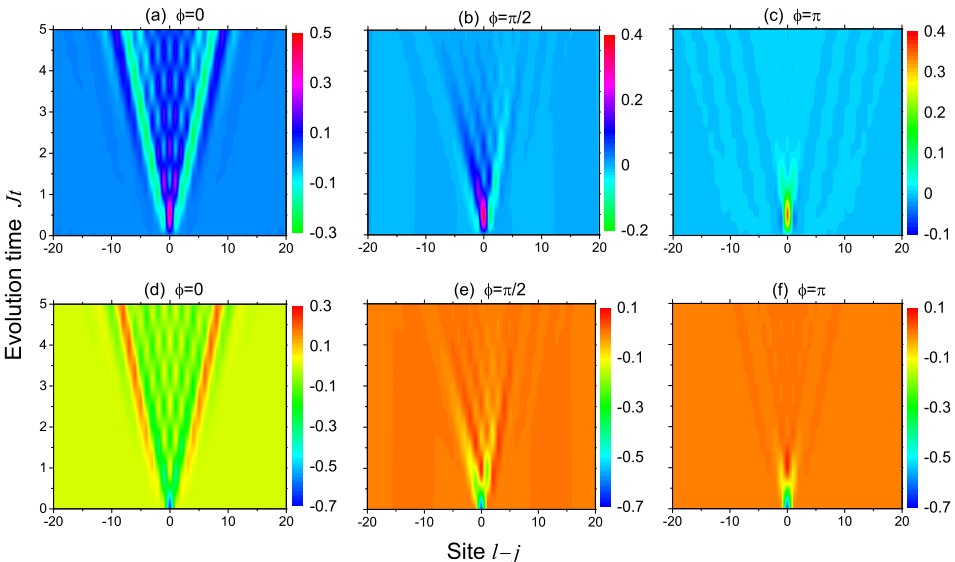

Figure 2: The real (top panel) and imaginary (bottom panel) part of the greater Green's function $G_{lj}^>(t)$ in a chain with $N = 65$ sites for three different statistical parameters $\phi = 0, \pi/2$ and $\pi$. The site $j$ is fixed at the center of the chain, $j = 33$, and $\mu/J = 2.0, \Delta/J = 0.1, \gamma_-/J = 0.1, \gamma_+/J = 0.05$.

When $t = 0$, the lesser Green's function would reduce to the steady-state one-particle density matrix, which is studied in Appendix.D. When $t \neq 0$, these Green's functions tell us the dynamical propagation of a single-particle excitation in space-time. After Fourier transformation, they can also give us the spectral functions, which are very important quantities in both theoretical and experimental studies.

In Fig.2 we plot the real and imaginary part the greater Green's function $G_{lj}^>(t)$ in a chain with $N = 65$ sites for three different statistical parameters $\phi = 0, \pi/2$ and $\pi$. The site $j$ is fixed at the center of the chain and the figure displays the propagation of the excitation in space-time. Spatial symmetry and temporal damping behaviors can be seen clearly. For $\phi = 0$, i.e., spinless fermions, the propagation shows a clear symmetric light cone. However, for $0 < \phi < \pi$, the light-cone becomes asymmetric, as shown in Fig.2(b) and Fig.2(e) for $\phi = \pi/2$. This asymmetric propagation is induced by the statistical parameter, since the Hamiltonian and the dissipators are symmetric under the spatial reflection about the chain center. To show this, we label the Green's function $G_{lj}^>(t)$ with the parameter $\phi$. Then we have

$$G_{lj}^>(t; \phi) = G_{l'j'}^>(t; -\phi), \tag{35}$$

where $l'(j')$ is the site that $l(j)$ is mapped to under reflection about the center of the chain. So the light-cones in Fig.2 should be symmetric only for $\phi = 0, \pi$. We stress that this symmetry holds only for symmetric Hamiltonian and dissipators as set in this paper. Asymmetric dissipations may also induce asymmetric light-cones even for $\phi = 0$ and $\pi$, as observed elsewhere [57].

We also observe that the greater Green's function decay rapidly for large statistical parameters. This behavior could be seen clearly in Fig.3, where the local Green's function $G_{jj}^>(t)$ at the center of the chain is plotted as a function of time for $\phi = 0, \pi/5, \pi/2$ and $\pi$. We see that in all cases $G_{jj}^>(t)$ oscillates and decays. The oscillation is a feature of the coherent Hamiltonian dynamics while the decay has two sources: (i) the boundary dissipations and (ii) the interactions between hard-core anyons. The dissipations can induce a finite (but small)

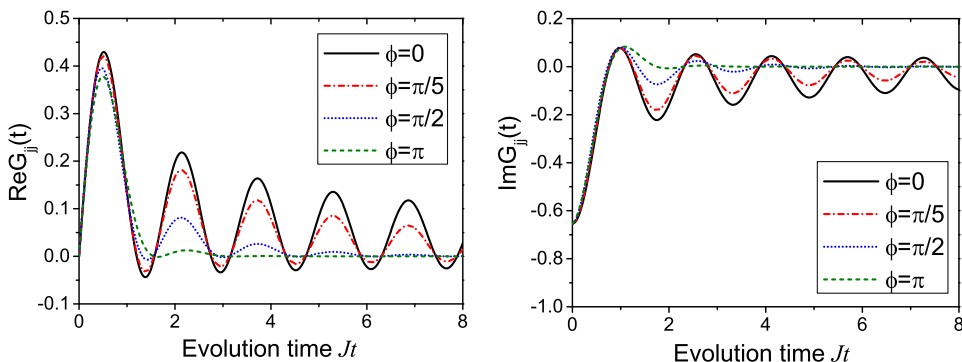

Figure 3: The real and imaginary part of the local greater Green's function $G_{jj}^>(t)$ at the center $j = 33$ in a chain with $N = 65$ sites for $\phi = 0, \pi/5, \pi/2$ and $\pi$. The other parameters are the same as that in Fig.2.

real part of the eigenvalues of $\mathbb{X}_+ + i\mathbb{H}$ [as shown in Fig.1], and hence all the corresponding modes decay with time. In addition, there exist strong effective interactions between the non-local excitations which would lead to scattering processes and finite relaxation rates. For the special case of $\phi = 0$, no interaction exists between the spinless fermions and hence the local Green's function decays slowly. However, as $\phi$ increases, the effective interaction grows, the relaxation rate becomes larger and larger, and hence $G_{jj}^>(t)$ decays more and more rapidly.

## 3.3 Full counting statistics of charge number

The charge number fluctuations in a subsystem is an important quantity in quantum many-body systems. It has been demonstrated that fluctuations and the full counting statistics (FCS) of charge or other conserved quantities (such as the block magnetization in certain spin chains) may contain information about the full entanglement scaling of a system split into two parts [93–96]. Here we consider the FCS of the charge distribution of a subsystem $A$ in the chain. For this purpose, we define the number operator $\hat{Q}_A$ as $\hat{Q}_A = \sum_{j \in A} \hat{c}_j^\dagger \hat{c}_j$, and a diagonal $N \times N$ matrix $\mathbb{D}_A$ with diagonal elements

$$(\mathbb{D}_A)_{jj} = \begin{cases} 1 & \text{if } j \in A, \\ 0 & \text{otherwise}. \end{cases}$$

Then $e^{\lambda \hat{Q}_A} = \hat{\Gamma}_1(\lambda \mathbb{D}_A) = \hat{\Gamma}_2(\lambda \tau_z \mathbb{D}_A) e^{\lambda \text{Tr}(\mathbb{D}_A)/2}$, which can be taken as a special Gaussian operator. Suppose that the initial state is a Gaussian state with the density matrix

$$\rho(0) = \frac{e^{-\beta \hat{H}_0}}{\text{Tr} e^{-\beta \hat{H}_0}}, \quad \hat{H}_0 = \frac{1}{2}(\hat{c}^\dagger, \hat{c}) \mathbb{H}_0 \begin{pmatrix} \hat{c} \\ \hat{c}^\dagger \end{pmatrix}.$$

In general the charge number in subsystem $A$ has no fixed value at time $t$; instead, it has a probability distribution. We would denote $P_n(t)$ as the probability that there are exactly $n$ charge in A at time $t$. Then the counting statistic function at time $t$ is

$$\chi(\lambda, t) = \sum_n P_n(t) e^{\lambda n} = \frac{1}{\text{Tr}[e^{-\beta \hat{H}_0}]} \text{Tr} \left\{ e^{\lambda \hat{Q}_A} e^{\mathcal{L}t} [e^{-\beta \hat{H}_0}] \right\}, \tag{36}$$

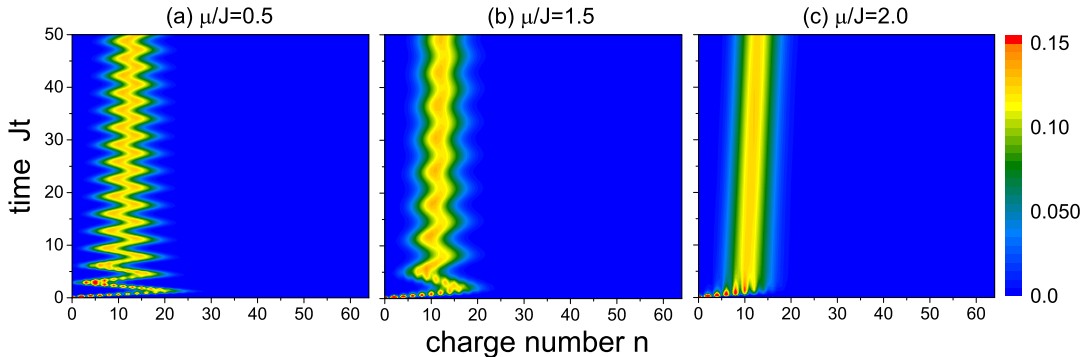

Figure 4: The dynamical evolution of the FCS $P_n(t)$ of the charge number in half of the chain from an initial vacuum state. The parameters are: $\Delta/J = 0.5, \gamma_-/J = 0.1, \gamma_+/J = 0.05$ and $N = 128$.

which could be taken as a special case of Eq.(21), and hence the result can be obtained immediately,

$$\chi(\lambda, t) = e^{\lambda \mathrm{Tr}(\mathbb{D}_A)/2} \sqrt{\det\left[\mathbb{B}(t) + e^{\lambda \tau_z \mathbb{D}_A}(\mathbb{1} - \mathbb{B}(t))\right]}, \tag{37}$$

where $\mathbb{B}(t) = \frac{1}{2}\mathbb{1} + \mathbb{Q}(t)\left(\mathbb{B}_0 - \frac{1}{2}\mathbb{1}\right)\bar{\mathbb{Q}}(t) + \mathbb{M}(t)$, and $\mathbb{B}_0 = [\mathbb{1} + e^{-\beta \mathbb{H}_0}]^{-1}$. This expression generalizes the result obtained by Klich [83] to dissipative systems. As $t \to \infty$, the state would approaches to the steady state with the density matrix $\rho_s = \sqrt{\det\left(\frac{1}{2}\mathbb{1} + \mathbb{M}_\infty\right)} \hat{\Gamma}_2(\mathbb{K}_0)$, and the counting statistic function approaches to its steady value

$$\chi_s(\lambda) = e^{\lambda \mathrm{Tr}(\mathbb{D}_A)/2} \sqrt{\det\left[\left(\frac{1}{2}\mathbb{1} + \mathbb{M}_\infty\right) + e^{\lambda \tau_z \mathbb{D}_A}\left(\frac{1}{2}\mathbb{1} - \mathbb{M}_\infty\right)\right]}. \tag{38}$$

From this expression of the counting statistic function we can derive the probability distribution $P_n$ of the charge number $\hat{Q}_A$.

In Fig.4 we plot the dynamical evolution of the FCS of the charge number in half of the chain with $N = 128$ sites. The initial state is chosen as the vacuum state, $\rho_0 = |0\rangle\langle 0|$, and hence at $t = 0$ we have $P_0 = 1, P_{n \neq 0} = 0$. As the system evolves, the distribution $P_n(t)$ changes with time. For $\mu = 0.5J < \mu_c$, the distribution $P_n(t)$ oscillates rapidly, while for $\mu = 2.0J > \mu_c$, the distribution almost does not oscillate and monotonically approaches to its steady-state value. This could be taken as a dynamical signature of the NQPT occurring at $\mu = \mu_c$. For the parameters chosen in Fig.4, the relaxation time is very long and hence we plot the steady-state value in Fig.5. The left plot shows the distribution $P_n$ as a function of $\mu$ while the right plot shows the distribution for three representative chemical potentials, $\mu = 0, \mu = 1.5J$ and $\mu = 3.0J$. We see that there are obvious singularities at $\mu = \pm\mu_c$ and $\mu = 0$, where NQPT occurs. So we conclude that both the dynamical evolution and the steady-state value of the FCS of the charge number could reveal the NQPT.

## 3.4 Loschmidt Echo and Dynamical Quantum Phase Transitions

One particularly interesting phenomenon in real-time dynamics of quantum many-body systems are DQPTs in the sense that an observable changes nonsmoothly at a critical time after a quench [97,98]. Since in many experiments the physical systems are subject to dissipation, it

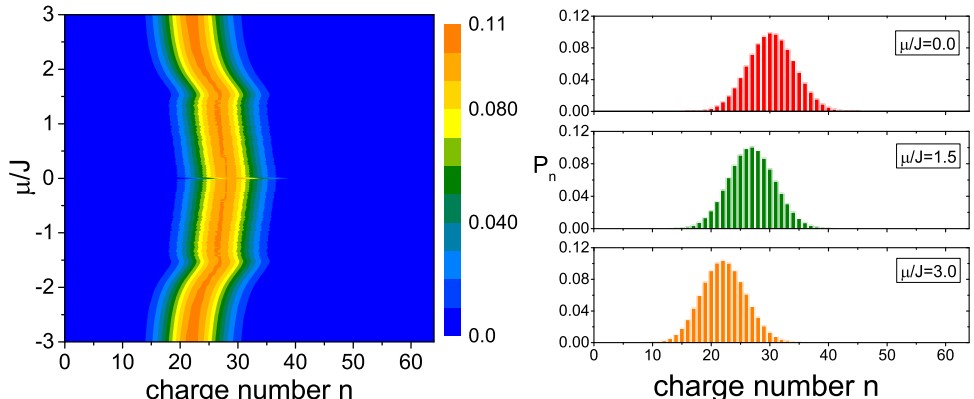

Figure 5: The steady-state FCS $P_n$ of the charge number in half of the chain. The parameters are the same as that in Fig.4. The left plot shows singularities at $\mu = 0$ and $\mu = \pm\mu_c = \pm 1.5J$.

is important to consider the fate of DQPTs in nonunitary dynamics. It has been shown that for simple Fermionic models the DQPTs may persist in the presence of dissipation [99–103]. Here we consider the possibility of DQPTs in the boundary-driven Kitatev chain. To characterize the quench dynamics we need a generalization of the Loschmidt echo $L(t)$ for mixed states. Following a recent Letter [103] we use the definition $L(t) = \text{Tr}[\rho(0)\rho(t)]$, and the rate function $r(t) = -(1/N)\ln L(t)$. As initial state we choose the vacuum state, which corresponds to the fully polarized ferromagnetic state in the context of the XY spin chain. This state can be taken as a Gaussian state with the density matrix $\rho = e^{-\beta\hat{H}_0}/\text{Tr}[e^{-\beta\hat{H}_0}]$, where $\hat{H}_0 = -\mu\sum_l \hat{c}_l^\dagger \hat{c}_l$ and $\beta\mu \to -\infty$. Then the Loschmidt echo $L(t)$ takes the form of Eq.(21) and can be simplified as

$$L(t) = \sqrt{\det\left[\mathbb{B}_0\mathbb{B} + (\mathbb{1} - \mathbb{B}_0)(\mathbb{1} - \mathbb{B})\right]}, \qquad (39)$$

and the rate function

$$r(t) = -\frac{1}{2N}\text{Tr}\ln\left[\mathbb{B}_0\mathbb{B} + (\mathbb{1} - \mathbb{B}_0)(\mathbb{1} - \mathbb{B})\right], \qquad (40)$$

where $\mathbb{B} = \frac{1}{2}\mathbb{1} + \mathbb{Q}(t)\left(\mathbb{B}_0 - \frac{1}{2}\mathbb{1}\right)\bar{\mathbb{Q}}(t) + \mathbb{M}(t)$ and $\mathbb{B}_0 = \left[\mathbb{1} + e^{-\beta\mathbb{H}_0}\right]^{-1}$.

In Fig.6 we show this rate function for several different dissipation rates and system sizes. We see that for the chosen parameters DQPTs occur, i.e., the rate function develops cusps at critical times. In the left plot we fix the dissipation rates $\gamma_{1\pm} = \gamma_{N\pm} = \gamma_{\pm}$. and increase the system size $N$. We see that the cusps are smoothed for small system sizes, but becomes sharper and sharper as the size increases. In the right plot we fix the system size $N = 100$ and increase the dissipation rates. It's obvious that the dissipations lead to a damping of the peaks but the cusps still persist. Even more interestingly, for the chosen parameters, a new cusp emerges near $Jt = 5$, where the unitary dynamics shows a plateau. The persistence of DQPTs and the emergence of new cusps in dissipative dynamics is generic and does not require fine turning of parameters. This can be easily verified numerically by using our theoretical approach.

## 4 Conclusion and discussion

In summary, we have developed a general theoretical approach to solve open fermion systems and apply it to systems with quadratic Lindbladian. We focus on the dynamical correlations

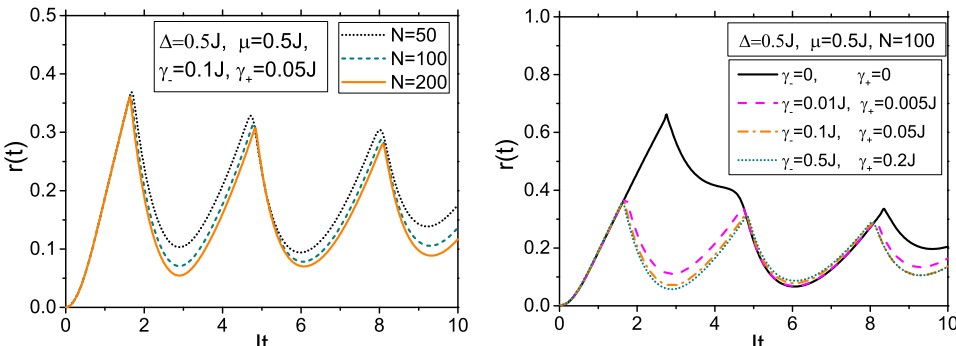

Figure 6: Loschmidt rate function $r(t)$ of the boundary-driven Kitaev chain. The dissipation rates are chosen to be $\gamma_{1\pm} = \gamma_{N\pm} = \gamma_{\pm}$. The left plot shows the rate function for fixed dissipation and different system sizes $N$. The right plot shows the rate function for fixed $N = 100$ and increasing dissipation rates.

of nonlocal operators and give exact explicit formulas based on our characteristic function approach. We then take the boundary-driven Kitaev chain as an example to illustrate the general ideas and formulas. We compute the Green's functions of hard-core anyons with statistical parameter $\phi$, and find that the propagation of the nonlocal excitations displays an asymmetric light-cone for $0 < \phi < \pi$, and the relaxation rate increases with $\phi$. In addition, two other types of nonlocal operator correlations such as the FCS of the charge number and the Loschmidt echo in quench dynamics are also analyzed and explicit formulas are obtained. The FCS shows clear signature of the steady-state NQPT, while the Loschmidt echo rate function exhibits cusps at some critical times in the quench from the vacuum state, demonstrating DQPTs in this dissipative system.

The characteristic function approach is a new and general theoretical method to treat open fermion systems. We would apply and extend this method to solve some other physical problems. For example, in the presence of dephasing, the Liouvillian is no longer quadratic and has no simple solutions like the quadratic Lindbladian. However, we find that the dynamical correlation functions can be obtained by making Taylor expansions of the characteristic function. Another important application is the full counting statistics in dissipative transport. Introduction of a counting field brings nonlocal operators naturally, which can be treated by using the techniques given in this paper. Results in these directions would be presented in future works.

## Acknowledgements

This work has been supported by the Fundamental Research Funds for the Provincial Universities of Zhejiang, Grant No.2021J014. We also acknowledge financial support from the Key Laboratory of Oceanographic Big Data Mining & Application of Zhejiang Province, Zhejiang Ocean University, Zhoushan, Zhejiang, China.

# A   Some useful formulas

In this appendix we give some concepts and formulas that are useful in deriving and understanding the results in the main text.

(1) The parity operator $\hat{P}_F$ in $\mathcal{K}$ can be defined by the transformation $\hat{P}_F(\hat{c}, \hat{c}^\dagger)\hat{P}_F = (-\hat{c}, -\hat{c}^\dagger)$. Obviously, one representation of the parity operator is $\hat{P}_F = e^{i\pi\hat{N}}$. Similarly, the parity operator $P_g$ in $\mathcal{G}$ can be defined as $P_g f(\bar{\xi}, \xi) = f(-\bar{\xi}, -\xi)$, and one representation of $P_g$ is

$$P_g = \exp\left[ i\pi \sum_k (\xi_k \partial_k + \bar{\xi}_k \bar{\partial}_k) \right]. \tag{41}$$

(2) The displacement operator $\hat{D}(\xi) \equiv e^{\hat{c}^\dagger \xi - \bar{\xi}\hat{c}}$ has the properties:

$$\mathrm{Tr}\hat{D}(\xi) = 2^N, \quad \mathrm{Tr}\left[ e^{i\pi\hat{N}}\hat{D}(\xi) \right] = \prod_{k=1}^N \xi_k \bar{\xi}_k, \tag{42}$$

and the integration is

$$\int d\bar{\xi} d\xi \, \hat{D}(\xi) = \frac{1}{2^N} e^{i\pi\hat{N}}, \tag{43}$$

where $\int d\bar{\xi}d\xi \equiv \int d\bar{\xi}_1 d\xi_1 d\bar{\xi}_2 d\xi_2 \cdots d\bar{\xi}_N d\xi_N$.

(3) A mixed operator involves both fermion operators and Grassmann variables, i.e., it's an element of the direct product space $\mathcal{K} \otimes \mathcal{G}$. Since fermion creation/annihilation operators anticommute with Grassmann variables, we should be careful in computing traces of such operators. We can use the following rules: (i) If $f(\bar{\eta}, \eta)$ has even parity, i.e., $f(\bar{\eta}, \eta) = f(-\bar{\eta}, -\eta)$, then $\mathrm{Tr}[\hat{A}f(\bar{\eta}, \eta)] = \mathrm{Tr}[\hat{A}]f(\bar{\eta}, \eta)$; (ii) If $f(\bar{\eta}, \eta)$ has odd parity, i.e., $f(\bar{\eta}, \eta) = -f(-\bar{\eta}, -\eta)$, then $\mathrm{Tr}[\hat{A}f(\bar{\eta}, \eta)] = \mathrm{Tr}[\hat{A}e^{i\pi\hat{N}}]f(\bar{\eta}, \eta)$.

(4) Here we give two basic Gaussian integrations for Grassmann variables. Denote $\alpha = (\alpha_1, \alpha_2, \cdots, \alpha_{2N})^T$ as a set of independent Grassmann variables and $Q$ a skew-symmetric matrix, then [80]

$$\int d\alpha_{2n} d\alpha_{2n-1} \cdots d\alpha_1 \, e^{\frac{1}{2}\alpha^T Q\alpha} = \mathrm{Pf}(Q), \tag{44}$$

where $\mathrm{Pf}(Q)$ denotes the Pfaffian of $Q$. Now suppose that $A$ is a $2N \times 2N$ matrix with the property $A + \tau_x A^T \tau_x = 0$, and $(\bar{\eta}, \eta) = (\bar{\eta}_1, \bar{\eta}_2, \cdots, \bar{\eta}_N, \eta_1, \eta_2, \cdots, \eta_n)$ is a $2N$-dimensional vector, then we can deduce the following integration from the above basic formula,

$$\int d\bar{\eta}d\eta \, \exp\left[ -\frac{1}{2}(\bar{\eta}, \eta)A\begin{pmatrix} \eta \\ \bar{\eta} \end{pmatrix} + (\bar{\xi}, \xi)\begin{pmatrix} \eta \\ \bar{\eta} \end{pmatrix} \right]$$
$$= \exp\left[ \frac{1}{2}\mathrm{Tr}\log(A\tau_z) \right]\exp\left[ -\frac{1}{2}(\bar{\xi}, \xi)A^{-1}\begin{pmatrix} \xi \\ \bar{\xi} \end{pmatrix} \right], \tag{45}$$

where $\int d\bar{\eta}d\eta \equiv \int d\bar{\eta}_1 d\eta_1 d\bar{\eta}_2 d\eta_2 \cdots d\bar{\eta}_N d\eta_N$. Note that one should make clear the order of the variables in making the integrations of Grassmann variables. Note also that the requirement $A + \tau_x A^T \tau_x = 0$ follows from the skew-symmetry property $Q + Q^T = 0$.

(5) We can also do "integration by parts" for functions of Grassmann variables. However, one should be careful about the anticommutation nature of Grassmann variables. Since $\partial_i[f(\xi)g(\xi)] = [\partial_i f(\xi)]g(\xi) + f(-\xi)\partial_i g(\xi)$, we have

$$\int d\xi_i [\partial_i f(\xi)]g(\xi) = -\int d\xi_i f(-\xi)\partial_i g(\xi). \tag{46}$$

(6) By defining the "Fourier kernal" $D(\xi|\eta) \equiv e^{\bar{\xi}\eta - \bar{\eta}\xi}$, we also have Fourier transformations in Grassmann algebra:

$$F(\bar{\xi}, \xi) = \int d\bar{\eta} d\eta \, D(\xi|\eta) f(\bar{\eta}, \eta), \quad f(\bar{\eta}, \eta) = \int d\bar{\xi} d\xi \, D(\eta|\xi) F(\bar{\xi}, \xi). \tag{47}$$

(7) The $\Theta$ mapping of basic Gaussian operators:

$$\text{Tr}\left[\hat{\Gamma}_2(\mathbb{K})\hat{D}(\xi)\right] = \sqrt{\det(\mathbb{1} + e^{\mathbb{K}})} \exp\left[-\frac{1}{2}(\bar{\xi}, \xi)\frac{1}{\mathbb{1} + e^{\mathbb{K}}}\left(\begin{array}{c} \xi \\ \bar{\xi} \end{array}\right)\right], \tag{48}$$

$$\text{Tr}\left[(\hat{c}^\dagger, \hat{c})\hat{\Gamma}_2(\mathbb{K})e^{i\pi\hat{N}}\hat{D}(\xi)\right] = -\left\{(\bar{\xi}, \xi)\frac{1}{\mathbb{1} + e^{\mathbb{K}}}\right\}\text{Tr}\left[\hat{\Gamma}_2(\mathbb{K})\hat{D}(\xi)\right], \tag{49}$$

where $\mathbb{K} + \tau_x \mathbb{K}^T \tau_x = 0$ is required.

(8) The $\Omega$ mapping of basic Gaussian functions:

$$\Omega\left\{\exp\left[-\frac{1}{2}(\bar{\xi}, \xi)\mathbb{B}\left(\begin{array}{c} \xi \\ \bar{\xi} \end{array}\right)\right]\right\} = \sqrt{\det \mathbb{B}} \, \hat{\Gamma}_2(\mathbb{K}), \tag{50}$$

$$\Omega\left\{(\bar{\xi}, \xi)\exp\left[-\frac{1}{2}(\bar{\xi}, \xi)\mathbb{B}\left(\begin{array}{c} \xi \\ \bar{\xi} \end{array}\right)\right]\right\} = -(\hat{c}^\dagger, \hat{c})\frac{\sqrt{\det \mathbb{B}}}{\mathbb{B}}\hat{\Gamma}_2(\mathbb{K})e^{i\pi\hat{N}}, \tag{51}$$

where $\mathbb{B}(\mathbb{1} + e^{\mathbb{K}}) = \mathbb{1}$ satisfies the relation $\mathbb{B} + \tau_x \mathbb{B}^T \tau_x = \mathbb{1}$, while the matrix $\mathbb{K}$ satisfies $\mathbb{K} + \tau_x \mathbb{K}^T \tau_x = 0$.

(9) As stated in the main text, we can make analogy with concepts in quantum optics and define some phase-space functions such as the $Q$-function or $P$-function. Investigations along this line deserve further systematic studies. Here we just give some preliminary results about the $Q$-function. For any operator $\hat{A}$, its $Q$-function can be defined as

$$A_Q(\bar{\xi}, \xi) \equiv \frac{\langle\xi|\hat{A}|\xi\rangle}{\langle\xi|\xi\rangle}, \tag{52}$$

where $|\xi\rangle$ is the fermionic coherent state. The $Q$-function is related with the characteristic function $A_C$ by a proper Fourier transformation. However, we stress again that one should be careful about the anticommutation nature of Grassmann variables. Here we should distinguish the different parities of the functions/operators defined above in this Appendix. For even-parity functions,

$$A_Q(\bar{\xi}, -\xi) = e^{2\bar{\xi}\xi} \int d\bar{\eta} d\eta \, e^{-\bar{\eta}\eta/2} A_C(\bar{\eta}, \eta) D(\eta|\xi), \tag{53}$$

while for odd-parity functions,

$$A_Q(\bar{\xi}, \xi) = \int d\bar{\eta} d\eta \, e^{-\bar{\eta}\eta/2} A_C(\bar{\eta}, \eta) D(\eta|\xi). \tag{54}$$

We will not give proof for these transformations here since (i) the proof is a little lengthy and (ii) the $Q$-function is not used in this paper. We just point out a future development direction of the characteristic function approach.

## B  Equation of Motion for the Characteristic Function

Here we sketch the derivation of the equation of motion for the characteristic function $F(\bar{\xi}, \xi)$ given by Eq.(7). We note that

$$(\hat{c}^\dagger, \hat{c})\mathbb{A}\left(\begin{array}{c} \xi \\ \bar{\xi} \end{array}\right) = -(\xi, \bar{\xi})\mathbb{A}^T\left(\begin{array}{c} \hat{c}^\dagger \\ \hat{c} \end{array}\right) = -(\bar{\xi}, \xi)\tau_x \mathbb{A}^T \tau_x \left(\begin{array}{c} \hat{c} \\ \hat{c}^\dagger \end{array}\right).$$

Then the displacement operator $\hat{D}(\xi) \equiv e^{\hat{c}^\dagger \xi - \bar{\xi}\hat{c}}$ has the following properties:

$$
\begin{aligned}
[\hat{D}(\xi), \hat{H}] &= \left[\hat{D}(\xi)\hat{H}\hat{D}^\dagger(\xi) - \hat{H}\right]\hat{D}(\xi) \\
&= \left[\frac{1}{2}(\hat{c}^\dagger - \bar{\xi}, \hat{c} - \xi)\mathbb{H}\begin{pmatrix} \hat{c} - \xi \\ \hat{c}^\dagger - \bar{\xi} \end{pmatrix} - \frac{1}{2}(\hat{c}^\dagger, \hat{c})\mathbb{H}\begin{pmatrix} \hat{c} \\ \hat{c}^\dagger \end{pmatrix}\right]\hat{D}(\xi) \\
&= \left[-\frac{1}{2}(\hat{c}^\dagger, \hat{c})\mathbb{H}\begin{pmatrix} \xi \\ \bar{\xi} \end{pmatrix} - \frac{1}{2}(\bar{\xi}, \xi)\mathbb{H}\begin{pmatrix} \hat{c} \\ \hat{c}^\dagger \end{pmatrix} + \frac{1}{2}(\bar{\xi}, \xi)\mathbb{H}\begin{pmatrix} \xi \\ \bar{\xi} \end{pmatrix}\right]\hat{D}(\xi) \\
&= \left[-\frac{1}{2}(\bar{\xi}, \xi)(\mathbb{H} - \tau_x \mathbb{H}^T \tau_x)\begin{pmatrix} \hat{c} \\ \hat{c}^\dagger \end{pmatrix} + \frac{1}{2}(\bar{\xi}, \xi)\mathbb{H}\begin{pmatrix} \xi \\ \bar{\xi} \end{pmatrix}\right]\hat{D}(\xi) \\
&= \left[-(\bar{\xi}, \xi)\mathbb{H}\begin{pmatrix} \hat{c} \\ \hat{c}^\dagger \end{pmatrix} + \frac{1}{2}(\bar{\xi}, \xi)\mathbb{H}\begin{pmatrix} \xi \\ \bar{\xi} \end{pmatrix}\right]\hat{D}(\xi) \\
&= \left[-(\bar{\xi}, \xi)\mathbb{H}\begin{pmatrix} \xi/2 - \bar{\partial} \\ \bar{\xi}/2 - \partial \end{pmatrix} + \frac{1}{2}(\bar{\xi}, \xi)\mathbb{H}\begin{pmatrix} \xi \\ \bar{\xi} \end{pmatrix}\right]\hat{D}(\xi) \\
&= (\bar{\xi}, \xi)\mathbb{H}\begin{pmatrix} \bar{\partial} \\ \partial \end{pmatrix}\hat{D}(\xi),
\end{aligned}
$$

and

$$
\begin{aligned}
&2\hat{L}_\mu^\dagger \hat{D}(\xi)\hat{L}_\mu - \hat{L}_\mu^\dagger \hat{L}_\mu \hat{D}(\xi) - \hat{D}(\xi)\hat{L}_\mu^\dagger \hat{L}_\mu \\
&= \left[2\hat{L}_\mu^\dagger \hat{D}(\xi)\hat{L}_\mu \hat{D}^\dagger(\xi) - \hat{L}_\mu^\dagger \hat{L}_\mu - \hat{D}(\xi)\hat{L}_\mu^\dagger \hat{L}_\mu \hat{D}^\dagger(\xi)\right]\hat{D}(\xi) \\
&= \left[(\bar{\xi}, \xi)L_\mu L_\mu^\dagger\begin{pmatrix} \hat{c} \\ \hat{c}^\dagger \end{pmatrix} - (\hat{c}^\dagger, \hat{c})L_\mu L_\mu^\dagger\begin{pmatrix} \xi \\ \bar{\xi} \end{pmatrix} - (\bar{\xi}, \xi)L_\mu L_\mu^\dagger\begin{pmatrix} \xi \\ \bar{\xi} \end{pmatrix}\right]\hat{D}(\xi) \\
&= \left[(\bar{\xi}, \xi)\mathbb{X}_+\begin{pmatrix} \hat{c} \\ \hat{c}^\dagger \end{pmatrix} - (\bar{\xi}, \xi)L_\mu L_\mu^\dagger\begin{pmatrix} \xi \\ \bar{\xi} \end{pmatrix}\right]\hat{D}(\xi) \\
&= \left[(\bar{\xi}, \xi)\mathbb{X}_+\begin{pmatrix} \xi/2 - \bar{\partial} \\ \bar{\xi}/2 - \partial \end{pmatrix} - (\bar{\xi}, \xi)L_\mu L_\mu^\dagger\begin{pmatrix} \xi \\ \bar{\xi} \end{pmatrix}\right]\hat{D}(\xi) \\
&= \left[-(\bar{\xi}, \xi)\mathbb{X}_+\begin{pmatrix} \bar{\partial} \\ \partial \end{pmatrix} - \frac{1}{2}(\bar{\xi}, \xi)\mathbb{X}_-\begin{pmatrix} \xi \\ \bar{\xi} \end{pmatrix}\right]\hat{D}(\xi),
\end{aligned}
$$

where $\mathbb{X}_\pm$ is defined by Eq.(8). The equation of motion for $F(\bar{\xi}, \xi)$ reads

$$
\partial_t F = \text{Tr}\left[\mathcal{L}(\rho)\hat{D}(\xi)\right] = \text{Tr}\left\{\rho \mathcal{L}_{\text{ad}}[\hat{D}(\xi)]\right\},
$$

where $\mathcal{L}_{\text{ad}}$ is the adjoint superoperator of $\mathcal{L}$,

$$
\mathcal{L}_{\text{ad}}[\hat{D}(\xi)] = -i[\hat{D}(\xi), \hat{H}] + \sum_\mu \left[2\hat{L}_\mu^\dagger \hat{D}(\xi)\hat{L}_\mu - \hat{L}_\mu^\dagger \hat{L}_\mu \hat{D}(\xi) - \hat{D}(\xi)\hat{L}_\mu^\dagger \hat{L}_\mu\right].
$$

Inserting the expressions for $[\hat{D}(\xi), \hat{H}]$ and $[2\hat{L}_\mu^\dagger \hat{D}(\xi)\hat{L}_\mu - \hat{L}_\mu^\dagger \hat{L}_\mu \hat{D}(\xi) - \hat{D}(\xi)\hat{L}_\mu^\dagger \hat{L}_\mu]$ into this equation of motion leads to the final result, Eq.(7). In fact, the operator $\hat{D}(\xi)$ satisfies the same differential equation and hence its dynamical evolution can be written as

$$
\hat{D}(\bar{\xi}, \xi; t) = \hat{D}\left[(\bar{\xi}, \xi)\mathbb{Q}(t)\right]\exp\left[-\frac{1}{2}(\bar{\xi}, \xi)\mathbb{M}(t)\begin{pmatrix} \xi \\ \bar{\xi} \end{pmatrix}\right].
$$

Similar results have been obtained for bosonic operators [81].

# C  The sign problem of the Green's function

The conventional dissipation superoperator $\mathcal{D}$ with Lindblad operator $\hat{L}, \hat{L}^\dagger$ reads

$$\mathcal{D}[\circ] = 2\hat{L} \circ \hat{L}^\dagger - \left\{\hat{L}^\dagger \hat{L}, \circ\right\}. \tag{55}$$

However, if both the operator $\circ$ and the Lindblad operator $\hat{L}^{(\dagger)}$ are fermionic operators, i.e., they have odd Fermion number parity, then the dissipation superoperator should differ from the above one by having a minus sign in front of the $2\hat{L} \circ \hat{L}^\dagger$ term, leading to a new superoperator [82]:

$$\mathcal{D}_f[\circ] = -2\hat{L} \circ \hat{L}^\dagger - \left\{\hat{L}^\dagger \hat{L}, \circ\right\}. \tag{56}$$

This difference is due to the anticommutation nature of fermionic operators and has been proved from first principle [82]. However, we should note that these two superoperators are intimately connected: If $\hat{P}_F \hat{L} \hat{P}_F = -\hat{L}$, then

$$\hat{P}_F \mathcal{D}_f[\hat{P}_F \circ] = \mathcal{D}[\circ], \quad \hat{P}_F e^{\mathcal{D}_f t}[\hat{P}_F \circ] = e^{\mathcal{D} t}[\circ]. \tag{57}$$

Similarly,

$$\mathcal{D}_f[\circ \hat{P}_F]\hat{P}_F = \mathcal{D}[\circ], \quad e^{\mathcal{D}_f t}[\circ \hat{P}_F]\hat{P}_F = e^{\mathcal{D} t}[\circ]. \tag{58}$$

The proof is straightforward:
(1)

$$\begin{aligned}
\hat{P}_F \mathcal{D}_f[\hat{P}_F \circ] &= -2\hat{P}_F L\hat{P}_F \circ L^\dagger - \hat{P}_F\left\{L^\dagger L, \hat{P}_F \circ\right\} \\
&= 2L \circ L^\dagger - \left\{L^\dagger L, \circ\right\} = \mathcal{D}[\circ].
\end{aligned}$$

(2) Define $\tilde{A}(t) = \hat{P}_F e^{\mathcal{D}_f t}[\hat{P}_F A]$, and $A(t) = e^{\mathcal{D} t}[A]$, then

$$\frac{\partial}{\partial t}\tilde{A}(t) = \hat{P}_F \mathcal{D}_f\left\{e^{\mathcal{D}_f t}[\hat{P}_F A]\right\} = \hat{P}_F \mathcal{D}_f\left\{\hat{P}_F \hat{P}_F e^{\mathcal{D}_f t}[\hat{P}_F A]\right\} = \mathcal{D}[\tilde{A}(t)],$$

with the initial condition $\tilde{A}(t = 0) = A$. On the other hand, $A(t)$ satisfies the equation

$$\frac{\partial}{\partial t}A(t) = \mathcal{D}[A(t)],$$

with the initial condition $A(t = 0) = A$. So we see that $\tilde{A}(t)$ and $A(t)$ satisfy the same equation of motion and the same initial condition, and hence $\tilde{A}(t) = A(t)$, i.e.,

$$\hat{P}_F e^{\mathcal{D}_f t}[\hat{P}_F \circ] = e^{\mathcal{D} t}[\circ].$$

Similarly we can prove the other equations.

# D  Steady State and Static Correlations

The dynamical correlation functions would reduce to static ones just by taking the evolution time $t = 0$. Therefore our formalism is also useful for computing static correlations of local or nonlocal excitations. This special limiting case is nontrivial since the correlation functions may be used to detect the NQPT. In addition, they can also be used to test the numerical computation codes for the more complicated dynamical correlations. Here we study the static correlation functions in the steady state. We first give the explicit expression of the steady state characteristic function, and then study the the momentum distribution of anyons, which shows clear signatures of the NQPT.

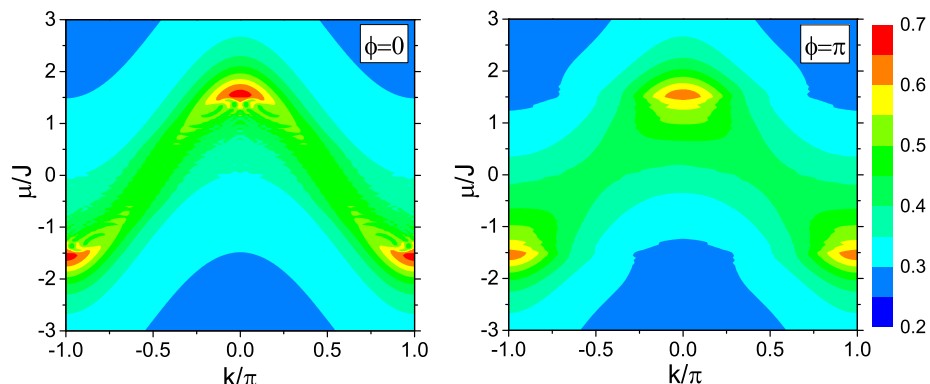

Figure 7: The $k$-distribution $n(k)$ in the steady state with the statistical parameter $\phi = 0$ (left) and $\phi = \pi$ (right). The other parameters $\Delta, \gamma_\pm$ and $N$ are the same as in Fig.1. The critical chemical potential is $\mu_c/J = \pm 1.5$.

Suppose that the non-Hermitian matrix $\mathbb{X}_+ + i\mathbb{H}$ has the spectral decomposition

$$\mathbb{X}_+ + i\mathbb{H} = \sum_{k=1}^{2N} \lambda_k |\varphi_k^R\rangle\langle\varphi_k^L|,$$

where $\{\lambda_k\}$ are the eigenvalues and $\{|\varphi_k^{R(L)}\rangle\}$ the right (left) eigenvectors of $\mathbb{X}_+ + i\mathbb{H}$, satisfying the biorthonormal condition $\langle\varphi_k^L|\varphi_q^R\rangle = \delta_{k,q}$. We can prove that $\mathrm{Re}\lambda_k \geq 0$ for all $k$. For the boundary-driven Kitaev chain with a finite size $N$, we can numerically verify that $\mathrm{Re}\lambda_k > 0$ for all $k$. Then the steady state characteristic function is given by Eq.(12) with

$$\mathbb{M}_\infty = \sum_{m,n} \frac{\langle\varphi_m^L|\mathbb{X}_-|\varphi_n^L\rangle}{\lambda_m + \lambda_n^*} |\varphi_m^R\rangle\langle\varphi_n^R|. \tag{59}$$

Here we focus on the momentum distribution of anyons defined as [104]

$$n(k) \equiv \frac{1}{N} \sum_{j,l=1}^{N} e^{ik(j-l)} \langle\hat{f}_j^\dagger \hat{f}_l\rangle.$$

Such correlation functions of nonlocal operators can be computed by takeing the $t = 0$ limit of the lesser Green's function. In Fig.7 we plot this distribution for two statistical parameters $\phi = 0$ and $\phi = \pi$. We see that the behavior of $n(k)$ is qualitatively the same for different statistical parameters. When $|\mu| < |\mu_c|$, the $k$-distribution shows two maximums at $k \neq 0, \pi$, otherwise it shows only one maximum at $k = 0$ or $\pi$. So the NQPT occurring at $\mu_c$ can be clearly characterized by the $k$-distribution function.

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
