# Peer review of "Exact dynamical correlations of nonlocal operators in quadratic open Fermion systems: a characteristic function approach"

_SciPost Physics Core, doi:SciPost Phys. Core 5, 027 (2022)_

## Round 1 · Referee Report · Anonymous (Referee 1) · 2022-2-10

Report

This work introduces the interesting characteristic function approach as an alternative to the usual way to treat quadratic open systems with linear dissipation. I believe this approach can be sometimes more elegant, but I expect it can produce results exactly in the cases where the previously introduced approaches work. If this is not the case, I suggest the author to stress the difference more explicitly. More comments on this in the manuscript would be useful. Note that I didn’t check the derivation in all details, but I believe the author checked the results versus exact diagonalisation results and results from the literature.

In the second part, the method is applied to the “Kitaev chain”, where it reproduces the results on correlation functions and furthermore studies nonlocal excitations, which were not studied before. It also studies the full counting statistics and Loschmidt echo.

I believe this work contains publishable results, but it needs some work on clarity before it can be published in the appropriate journal, for which I provide suggestions below. Because of its technical nature, I suggest that after the modifications are made, the paper is published in SciPost Physics Core.
For me to recommend it to SciPost Physics, the paper would need to clearly demonstrate that the new method goes beyond the reach of the previous methods, and why these new results are important.
* * *
Some questions and suggestions for clarity:

Line 71-73: The author suggests the method is useful for generic open systems, but this claim is not supported by anything, as the method is used only for quadratic systems. I would suggest omitting the claim about the generality of the method or providing arguments.
Why is this method better than using only the commutation relations? Is there something it can do which can’t be done without it? Or is it only sometimes more convenient? Maybe comment on that at the end of section 2.1. Please comment also on differences to the ‘third quantisation method’: advantages, disadvantages, compare the sizes of the matrices…
How are the tools mentioned in Line 116-117 useful?

Can the method go beyond linear Lindbladians or quadratic Hamiltonians?

Function D(.|.) after Line 111 is not defined.

More explanation is needed around Eqs. 7 & 8.. Maybe expand with a paragraph or two, or add an appendix.

More clearly introduce the quantities of interest defined in Eq 19 & 20. There should be a definition that is clear and intuitive before writing the trace, which is hard to digest quickly ( something along the lines of <c_i(0) c_j(t)> ). Maybe also repeat how/where(which space) Tr acts.
Also, please explicitly define and introduce nonlocal single-particle operators.

Line 176: I propose to comment on the parity issue here and define the relevant objects in the main text. Expecting the reader to go to the appendix for a definition of the object is unreasonable.

L215: I don’t understand why the name Kitaev chain is appropriate here, as the model was studied before Kitaev.

L222 Author uses a very specific symmetric dissipation on only two sites. I would expect a comment that this is a very special setting. Also later it should be clear that the results are for this specific setting.

L259: The motivation is not very convincing, maybe rephasing would help. Also at the end of Sec 3.2 it is not clear why the results in Fig.2 are relevant/useful/interesting.

L280: Stress that the light-cone is symmetric at phi=0 only for symmetric dissipation. Different dissipation at the left/right end of the chain produces asymmetric light-cones, as was discussed in the literature before (which is cited by this work).

After L300 there is a typo in the equation.

L303 Please define P_n(t)and better introduce FCS. What exactly does P_n(t) tell us for a specific n?

L401-402 Why it should differ? Explain.

L414 Why App C? Explain why it is here, what it contains, why is it relevant.

  • validity: high
  • significance: ok
  • originality: ok
  • clarity: ok
  • formatting: excellent
  • grammar: good

Author:  Qing-Wei Wang  on 2022-03-23  [id 2313]

(in reply to Report 1 on 2022-02-10)
Category:
answer to question

First of all, we thank the referee for the useful comments on this paper.

(1)Line 71-73: The method is general and can go beyond quadratic open systems since the key mappings defined in Sec.2.1 are independent of the concrete form the system Hamiltonian and dissipators. Further applications to other systems beyond quadratic Lindbladian would be presented in our future works, as discussed in the last paragraph of Sec.4. This claim is not supported by illustrating example in this paper, but the generality of our method (mainly refer to the mappings, but not the exact solutions) is also obvious. We add some arguments in Sec.2.1 to make this point clearer.

(2)We comment on the advantage and disadvantage of our method by comparing with the third quantization’’ method at the end of Sec.2.1

(3)The tools mentioned in Line 116-117: We add some comments on the usage of the tools.

(4)The mappings between the Liouville-Fock space and the Grassmann algebra are general and have nothing to do with the concrete form of the Hamiltonian and Lindbladian. The master equation can always be transformed to a partial differential equation of the characteristic function. However, we can’t always solve this equation exactly. It’s the exact solution that is restricted to quadratic open systems, but not the method; the method is general if we allow approximations such as perturbation expansion.

(5)Line112: The definition of D(.|.) is added.

(6)An appendix is added to explain Eqs.(7) and (8) in more details. See Appendix.B in the revised version.

(7)Line173-175: We present some introduction of the quantities defined in Eq.(19) and (20), and a definition of nonlocal single-particle operators.

(8)Line176: We add some comment on the parity issue.

(9)Line215: The name ‘Kitaev chain’ is widely used in the literature. For example, A. Carmele et al., Phys. Rev. B 92, 195107 (2015); G.Y.Chitov, Phys.Rev.B 97, 085131 (2018); S. Bandyopadhyay et al., Phys. Rev.B 101, 104307 (2020); Dan-Bo Zhang et al., Phys. Rev. Lett. 127, 020502 (2021), and references therein. In my understanding, the model is named after Kitaev since Kitaev’s work on topological superconductivity [A. Kitaev, Phys.-Usp. 44, 131 (2001)] stimulated vast interests in this model.

(10)Line222: Some comments on the symmetric dissipation setting are added at the end of this paragraph.

(11)Line259: The motivation is rewritten. More discussions on Fig.2 is added.

(12)Line280: We add the stress that the light-cone is symmetric for ϕ=0 only for symmetric dissipation.

(13)Line300: The typo is fixed. Thank you very much for pointing out this typo.

(14)Line303: The FCS Pn(t) is introduced more clearly. It tells us the probability that there are exactly n charge in the subsystem A at time t.

(15)Line401-402: We add some explanations on the difference.

(16)Line414: We explain the relevance and the contents of App.C at the beginning of this appendix.

Author:  Qing-Wei Wang  on 2022-03-23  [id 2312]

(in reply to Report 1 on 2022-02-10)
Category:
answer to question

First of all, we thank the referee for the useful comments on this paper.

(1)Line 71-73: The method is general and can go beyond quadratic open systems since the key mappings defined in Sec.2.1 are independent of the concrete form the system Hamiltonian and dissipators. Further applications to other systems beyond quadratic Lindbladian would be presented in our future works, as discussed in the last paragraph of Sec.4. This claim is not supported by illustrating example in this paper, but the generality of our method (mainly refer to the mappings, but not the exact solutions) is also obvious. We add some arguments in Sec.2.1 to make this point clearer.

(2)We comment on the advantage and disadvantage of our method by comparing with the third quantization’’ method at the end of Sec.2.1

(3)The tools mentioned in Line 116-117: We add some comments on the usage of the tools.

(4)The mappings between the Liouville-Fock space and the Grassmann algebra are general and have nothing to do with the concrete form of the Hamiltonian and Lindbladian. The master equation can always be transformed to a partial differential equation of the characteristic function. However, we can’t always solve this equation exactly. It’s the exact solution that is restricted to quadratic open systems, but not the method; the method is general if we allow approximations such as perturbation expansion.

(5)Line112: The definition of D(.|.) is added.

(6)An appendix is added to explain Eqs.(7) and (8) in more details. See Appendix.B in the revised version.

(7)Line173-175: We present some introduction of the quantities defined in Eq.(19) and (20), and a definition of nonlocal single-particle operators.

(8)Line176: We add some comment on the parity issue.

(9)Line215: The name ‘Kitaev chain’ is widely used in the literature. For example, A. Carmele et al., Phys. Rev. B 92, 195107 (2015); G.Y.Chitov, Phys.Rev.B 97, 085131 (2018); S. Bandyopadhyay et al., Phys. Rev.B 101, 104307 (2020); Dan-Bo Zhang et al., Phys. Rev. Lett. 127, 020502 (2021), and references therein. In my understanding, the model is named after Kitaev since Kitaev’s work on topological superconductivity [A. Kitaev, Phys.-Usp. 44, 131 (2001)] stimulated vast interests in this model.

(10)Line222: Some comments on the symmetric dissipation setting are added at the end of this paragraph.

(11)Line259: The motivation is rewritten. More discussions on Fig.2 is added.

(12)Line280: We add the stress that the light-cone is symmetric for ϕ=0 only for symmetric dissipation.

(13)Line300: The typo is fixed. Thank you very much for pointing out this typo.

(14)Line303: The FCS Pn(t) is introduced more clearly. It tells us the probability that there are exactly n charge in the subsystem A at time t.

(15)Line401-402: We add some explanations on the difference.

(16)Line414: We explain the relevance and the contents of App.C at the beginning of this appendix.

---

## Round 1 · Referee Report · Anonymous (Referee 2) · 2022-2-28

Strengths

- introduces a new method and shows its value and its applicability.
- considers applications of current interest in physics.

Weaknesses

- to my understanding, only applicable to Gaussian fermionic systems.

Report

I have read the manuscript by Wang. It introduces a formalism for open quadratic fermionic systems based on a characteristic function approach.
I find the idea very interesting. I have been using characteristic functions for bosonic systems for a long time and always felt I was missing a similar formalism for fermions.

The formalism developed is then applied to, in my opinion, interesting problems and I believe it will be useful also to other researchers working with open (quadratic) fermionic systems.
Of course, since the examples and the formalism belong to the realm of Gaussian systems, the same results could have been obtained with other methods. However, I find it remarkable that with the method of the author it is possible to straightforwardly compute quantities that would be almost impractical with standard techniques (e.g. Wick’s theorem).

In addition, I have to say that I very much like Eq. (9). (If such an equation already appeared somewhere else, the author should appropriately cite the reference).

In fact, I believe that the trace operation, which is implicit in F0, could be “undone”. In my opinion this is possible since Eq. 9 holds for any initial Gaussian ρ. (By the way, is it necessary to have a Gaussian initial state to prove Eq. 9?)
By undoing the trace operation, the author could write a dynamical equation for the operator D, that would read something like

Dt(ξ,ˉξ)=D(Q(t)(ξ,ˉξ))e1/2(ˉξ,ξ)M(t)(ξ,ˉξ).

While this is of course already contained in Eq. (9), I believe that reporting the above expression has its value since it would pair the known one for bosonic quasi-free semigroups [see e.g. the equation just before Eq. (1) in arxiv:0909.0408].

After the following comments have been taken into account by the author, my recommendation is to accept the paper for publication in Scipost.

Minor comments or suggestions:
1)
In the abstract I would not talk about the angle ϕ. I don’t think this is standard and may be confusing here.

2) Line 60/61: aren't “challenging” and “highly nontrivial” basically synonyms? I would just use one of the two adjectives here.

3)
Line 107: in the definition of D, the operator c and the variable ξ are used. These are not defined. I guess that c,c and ξ,ˉξ are vectors but it would be good to explicitly define them here. (For instance, c and c are implicitly defined only after Eq. 5)

4)
Eq. 9 is very similar to its counterpart for bosonic operators (see e.g. arxiv:0909.0408, or many other references). Maybe the author could mention this.

5)
In line 142-143, where the author says that the results of the section could be obtained by other methods, it would be good to add some references to these methods.

6)
In the vicinity of Eq. (14). Can the author also add the equation governing the time-evolution of the covariance matrix as obtained by Eq. 13 and not only the stationary one? This is straightforward, I would say, but it would be good to have it in the paper.

7)
Why is the constraint in line 171, K+τKTτ, needed? Is this just to get rid of an overall (possibly real phase) or does it have a deeper meaning? A comment would be helpful.

8) This is a suggestion that I leave to the author. In Fig. 1, there are shown the eigenvalues of HiX+, while in line 146 the matrix discussed is X++iH. Clearly one is i times the other, but I think it would be better to show in Fig. 1 the eigenvalues of X++iH. In this case, the caption should also be corrected inverting the role of real and imaginary parts.

9) With regards to the discussion between lines 228 and 239, I just wanted to note that, in arxiv:1805.10060, a boundary dissipation, only involving dephasing on a single boundary, was actually exploited to enhance the “lifetime” of the strong zero mode at the opposite boundary. Something similar should also be possible for linear jump operators. What is the author opinion on this?

  • validity: high
  • significance: good
  • originality: good
  • clarity: good
  • formatting: good
  • grammar: good

Author:  Qing-Wei Wang  on 2022-03-23  [id 2314]

(in reply to Report 2 on 2022-02-28)
Category:
answer to question

First of all, we thank the referee for the useful comments on this paper.

(0)By undoing the trace, we can also obtain the dynamical evolution of the operator ˆD. We mention this in the Appendix.B (in the revised version) after giving the details of the derivation for Eq.(7) and (8). Thank you for pointing out this interesting point.

The Eq.(9): It’s not necessary to have a Gaussian initial state. We stress this in the revised version.

(1)In the abstract we don’t talk about the angle ϕ anymore.

(2)Line60-61: We remove “challenging’’ and use only “highly nontrivial’’.

(3)The definitions for the operator ˆc and the variable ξ are added between Eq.(1) and Eq.(2).

(4)We mention the similarity to the bosonic counterpart and cite the paper 0909.0408 [i.e., Quantum Info. Comput.10, 619 (2010)]

(5)Line143: We add some references to these “other methods”.

(6)We add the equation of motion for covariance matrix between Eq.(13) and Eq.(14).

(7)We explain the reason for constraint below Eq.(18).

(8)Fig.1 is re-plotted. The new figure shows the eigenvalues of X++iH.

(9)We add some comments on the case of dissipation on a single boundary in Sec.3.1. The work in 1805.10060 is interesting and we cite this publication.

---

## Round 1 · Referee Report · Anonymous (Referee 3) · 2022-3-8

Report

I find this work by Wang is interesting. The author develops a characteristic function approach for the study of quadratic fermionic open quantum systems.
The work is unavoidably technical, but written quite clearly. There are also few typos which can be readily fixed during the editorial process.

The driving: I imagine that the authors used the same dissipative driving on the first and last site because they do not wish to consider a transport scenario which, for the purpose of the paper, complicates the problem beyond what is needed. But then my question is why both dissipators are needed, especially when considering steady state scenarios. I would think that one could remove the dissipators at one of the two edges? Am I wrong? In case I am not, maybe the author could comment on that.

Eqs.(7) and (8): I feel that the author could help/guide more the reader in getting to derive these equations. At the moment it seems a bit abrupt.

Page 2: the inline formula for \hat{O}_j is not clear to me. The sum seems to be on the j, because the operators have a j index, but then the outcome is not a function of l. Maybe the operators should have index l instead of j?

Similarly, Eq.(30) is not clear. Possibly the operators in the exponent should have a sub-index m instead of l

Fig.1: for the imaginary part, I would be more interested at the small values of lambda. My understanding is that, in fact, they can be used to detect phase transitions, which is one of the interests of this work. Maybe log(-Im{\lambda}) would show this more clearly?

Green functions: would it be possible to clarify whether the authors here is considering these correlators computed over the steady state?

In Fig.2 and 3 I would expect that the oscillations dynamics comes together a decay of the quantities as all the modes of the superoperator decay. Could the author clarify this when discussing the figures?

The author cites both the papers by Lindblad and by Gorini-Kossakovsky-Sudarshan. At the same time, I think that the author should refer to Eq.(3) as the GKSL equation.

I would write Re(\lmabda) and Im(\lambda) with parenthesis.

For perturbative methods for open quantum systems I would also cite:
Michel et al. Eur. Phys. J. B 42, 555 (2004)
Lenarcic et Prosen, Phys. Rev. E 91, 030103 (2015)
Znidaric, Phys. Rev. B 99, 035143 (2019)

For transitions driven by dissipation I would also cite:
Morrison and Parkins, Phys. Rev. Lett. 100, 040403 (2008)
Kessler et al., Phys. Rev. A 86, 012116 (2012)
Balachandran et al. Phys. Rev. Lett. 120, 200603 (2018)

Regarding ESQPT in system with dissipation coupled at the boundary, it could worth citing also:
Guo and Poletti, Phys. Rev. A 94, 033610 (2016)

To conclude I recommend the paper, after the due changes, for publication in Scipost, however as a Core.

  • validity: high
  • significance: good
  • originality: good
  • clarity: good
  • formatting: excellent
  • grammar: good

Author:  Qing-Wei Wang  on 2022-03-23  [id 2315]

(in reply to Report 3 on 2022-03-08)
Category:
answer to question

First of all, we thank the referee for the useful comments on this paper.

In the following, all line numbers and equation numbers refer to that in the old version of the paper.

(1). On the driving. The referee is right that the symmetric setting of the boundary dissipation is not necessary, and one can remove the dissipators at one edge. This does not change the steady state phase diagram qualitatively. However, this will change the spatial symmetry of, for example, the k-distribution n(k) which is studied in Appendix.C, and the Green’s functions in real space which is investigated in Sec.3.2. We add some comments on this point below equation (27).

(2)An appendix is added to sketch the derivation of Eqs.(7) and (8). See Appendix.B in the revised version.

(3)The inline formula for ˆOj: the index of the operators in the summation should be l instead of j. This is a typo. Similarly, there is a typo in Eq.(30). Thank you very much for pointing out these typos.

(4)Your suggestion is very good. The imaginary part can be used to give the Liouvillian gap and hence detect the dissipative phase transition. We re-plot Fig.1 and add some discussions about the Liouvillian gap in Sec.3.1.

(5)The Green’s functions are computed in the steady state. We add an explanation below the definition of the greater Green’s function.

(6)The oscillations decay as a result of (i) all the modes of X++iH decay, and (ii) the relaxation induced by the strong interactions between hard-core anyon excitations. We clarify this at the end of Sec.3.2.

(7)We rename the equation (3) as the GKSL equation.

(8)We rewrite Re(λ) and Im(λ) with parenthesis in Sec.3.1.

(9)The mentioned publications are cited in appropriate places.

---

## Round 2 · Referee Report · Anonymous (Referee 1) · 2022-3-23

Report

I appreciate that the author thoroughly answered all remarks and clarified all points. In my opinion, the paper has been nicely improved is now ready to be published.

As a small detail, I would claim that point (ii) in L186 is wrong. Also in the third quantisation, all information is stored in the 2N * 2N matrix analogous to X_+ +i H mentioned in the manuscript, due to the block upper triangular form of the 4N * 4N matrix in the third quantisation.

---

## Round 2 · Referee Report · Anonymous (Referee 3) · 2022-3-27

Report

I thank the authors for carefully considering all my comments and suggestions, and fixing the small issues. I can thus recommend it for publication as it is.

---

## Round 2 · Author Response

We have made revisions according to the comments of the three referees. We resubmit the revised version to SciPost Physics Core.

---

## Round 2 · List of Changes

Note that the positions listed below refer to that in the old version [revised version] of the paper.

  1. “statistical angle” is replaced by “statistical parameter” throughout the paper.

  2. L8-9 [L8-9]: “an asymmetric light cone induced by the statistical angle ϕ and an increasing relaxation rate with ϕ” is changed to “an asymmetric light cone induced by the anyon satistical parameter and an increasing relaxation rate with this parameter”.

3.L53 [L54]: A typo is fixed, i.e., $\hat{c}_j^\dag \hat{c}_j$ should be $\hat{c}_l^\dag \hat{c}_l$.

  1. L60-61 [L61-62]: “ a challenging and highly nontrivial theoretical problem” is changed to “a highly nontrivial theoretical problem”.

5.L107 [L108-110]: We add one sentence: ``Here we use the notations ...... ''

6.L112 [L114-115]: We add: ``where $D(\xi|\eta)\equiv e^{\bar{\xi}\eta-\bar{\eta}\xi}$ is the Grassmann analogy of the usual Fourier transformation kernel for complex variables''.

  1. L114-117 [L118-138]. This paragraph is expanded into two paragraphs.

  2. L119 [L140-141]: “quantum Lindblad master equation” is changed to “Gorini-Kossakorsky-Sudarshan-Lindblad (GKSL) equation”. And “Lindbladian” is changed to “Liouvillian”.

  3. L129 [L150]: “a nontrivial and challenging problem” is changed to “a challenging problem”.

  4. L133 [L154-159]: Three sentences are added: “See Appendix.B for the details of the derivation. We comment that for a general Liouvillian ... ... solved exactly by standard technique.”

  5. L133 [L159]: “The solution with initial condition...” is changed to “The solution with an arbitrary initial condition...”.

  6. L139 [L165-167]: One comment is added: “ We comment that the structure of the solution Eq.(9) is very similar to its bosonic counter part (see, for example, the work by T. Heinosaari et al. [81]).”.

  7. L139-140 [L168-199]: Two more paragraphs are added just before Sec.2.2.

  8. L143 [L203]: “by other methods” is changed to “by other methods[54-63]”.

  9. L153-154 [L213-215]: We add: ``From the equation for $F(\bar{\xi},\xi)$ ... ... denotes anticommutation relation.''

  10. L171 [L234-241]: We add: “We comment that the requirement ... ...otherwise the equation would be lengthy.”

  11. L173-176 [L242-257]: These lines are rewritten by adding more arguments.

  12. L216 [L297]: “the Kitaev chain” is changed to “the Kitaev chain[84]”.

  13. Fig.1 [Fig.1] is re-plotted and the caption is changed correspondingly.

  14. L222 [L303]: “The simplest nontrivial dissipations act only on the first and last site” is changed to “For simplicity of this illustrating example we take dissipations which act only on the first and last sites”.

  15. L227 [L308-318]: Some more arguments are added: “We remark that the symmetric dissipative driving ... ... to illustrate the general technique developed above.”

  16. L228-239 [L319-332]: This paragraph is rewritten in accordance with the change of Fig.1.

23.Eq.(28) [Eq.(28) ]: “Reλ” is changed to “Im(λ)”.

  1. Eq.(30) [Eq.(30) ]: A typo is fixed.

  2. L259-263 [L352-362]: These lines are rewritten.

  3. L269 [L368]: We add: “where the average ⟨·⟩ is taken in the steady state.”

  4. L275 [L375-378]: We add: ``When $t \neq 0$, these Green's functions tell us ... ... important quantities in both theoretical and experimental studies.''

  5. L279 [L382-383]: We add “Spatial symmetry and temporal damping behaviors can be seen clearly.”

  6. L282 [L385-387]: The sentence “This asymmetric propagation is induced by the statistical angle.” is changed to “This asymmetric propagation is induced by the statistical parameter, since the Hamiltonian and the dissipators are symmetric under the spatial reflection about the chain center.”

  7. L284 [L389]: “that l is mapped” should be “ that l(j) is mapped”.

  8. L285 [L390-393]: We add “We stress that this symmetry holds only for symmetric Hamiltonian and dissipators ... ... as observed elsewhere [57]. ”

  9. L289-292 [L397-406]: These lines are rewritten.

  10. L300-301 [L414-415 ]: A typo is fixed .

  11. L303 [L417-419]: We add “In general the charge number ... ... that there are exactly n charge in A at time t.”

  12. Appendix.A [Appendix.A] is expanded by including more formulas.

  13. One new appendix is added, i.e., Appendix.B in the revised version.

  14. L403-404 [L560-561]: One sentence is added: “This difference is due to .... ”.

  15. L404 [L561]: “We should note that...” is changed to “However, we should note that...”.

  16. L414-415 [L572-579]: One paragraph is added at the beginning of Appendix.C [Appendix.D]: “The dynamical correlation functions would reduce to static ones ...... which shows clear signatures of the NQPT.”

  17. The following 14 publications are added into the bibliography: • M.Michel, J. Gemmer, G. Mahler, Eur. Phys. J. B 42, 555 (2004). • Z. Lenarcic and T. Prosen, Phys. Rev. E 91, 030103(R) (2015). • Marko Znidaric, Phys. Rev. B 99, 035143 (2019). • S. Morrison and A. S. Parkins, Phys. Rev. Lett. 100, 040403(2008). • E. M. Kessler, G. Giedke, A. Imamoglu, S. F. Yelin, M. D. Lukin, and J. I. Cirac, Phys. Rev. A 86, 012116 (2012). • V. Balachandran, G. Benenti, E. Pereira, G. Casati, and Dario Poletti, Phys. Rev. Lett. 120, 200603 (2018). • Chu Guo and Dario Poletti, Phys. Rev. A 94, 033610 (2016). • K. Yamamoto, M. Nakagawa, N. Tsuji, M. Ueda, and N. Kawakami, Phys. Rev. Lett. 127, 055301 (2021). • W. Berdanier, J. Marino, and E. Altman, Phys. Rev. Lett. 123, 230604 (2019). • L. M. Vasiloiu, F. Carollo, and J. P. Garrahan, Phys. Rev. B 98, 094308 (2018). • T. Heinosaari, A. S. Holevo, M. M. Wolf, Quantum Inf. Comp. 10: 0619-0635 (2010). • A. Kitaev, Phys.-Usp. 44, 131 (2001). • Qing-Wei Wang, arXiv:2202.06543. • C. Itzykson, and J.-M. Drouffffe, Statistical Field Theory, Cambridge University Press, 1989.

---

## Editorial Decision

published